# Whole-genome landscape of mucosal melanoma reveals diverse drivers and therapeutic targets

Felicity Newell[1,14], Yan Kong[2,14], James S. Wilmott[3,4,14], Peter A. Johansson[1], Peter M. Ferguson [3,4], Chuanliang Cui[2], Zhongwu Li[2], Stephen H. Kazakoff[1], Hazel Burke[4], Tristan J. Dodds[3,4], Ann-Marie Patch [1], Katia Nones[1], Varsha Tembe[5], Ping Shang[4], Louise van der Weyden[6], Kim Wong [6], Oliver Holmes[1], Serigne Lo [3,4], Conrad Leonard[1], Scott Wood[1], Qinying Xu[1], Robert V. Rawson[3,7], Pamela Mukhopadhyay[1], Reinhard Dummer[8], Mitchell P. Levesque[8], Göran Jönsson[9], Xuan Wang[2], Iwei Yeh[10], Hong Wu[11], Nancy Joseph[12], Boris C. Bastian [10], Georgina V. Long[3,4,13], Andrew J. Spillane[3,4], Kerwin F. Shannon[3,4], John F. Thompson [3,4,7], Robyn P.M. Saw[3,4], David J. Adams[6], Lu Si[2], John V. Pearson[1], Nicholas K. Hayward[1,15], Nicola Waddell[1,15], Graham J. Mann[3,5,15], Jun Guo[2,15] & Richard A. Scolyer [3,4,7,15]

Knowledge of key drivers and therapeutic targets in mucosal melanoma is limited due to the paucity of comprehensive mutation data on this rare tumor type. To better understand the genomic landscape of mucosal melanoma, here we describe whole genome sequencing analysis of 67 tumors and validation of driver gene mutations by exome sequencing of 45 tumors. Tumors have a low point mutation burden and high numbers of structural variants, including recurrent structural rearrangements targeting *TERT*, *CDK4* and *MDM2*. Significantly mutated genes are *NRAS*, *BRAF*, *NF1*, *KIT*, *SF3B1*, *TP53*, *SPRED1*, *ATRX*, *HLA-A* and *CHD8*. *SF3B1* mutations occur more commonly in female genital and anorectal melanomas and *CTNNB1* mutations implicate a role for WNT signaling defects in the genesis of some mucosal melanomas. *TERT* aberrations and *ATRX* mutations are associated with alterations in telomere length. Mutation profiles of the majority of mucosal melanomas suggest potential susceptibility to CDK4/6 and/or MEK inhibitors.

[1] QIMR Berghofer Medical Research Institute, Brisbane, QLD 4006, Australia. [2] Department of Renal Cancer and Melanoma, Key Laboratory of Carcinogenesis and Translational Research, Ministry of Education, Peking University Cancer Hospital & Institute, Beijing 100142, China. [3] Melanoma Institute Australia, The University of Sydney, Sydney, NSW 2065, Australia. [4] Sydney Medical School, The University of Sydney, Sydney, NSW 2006, Australia. [5] Centre for Cancer Research, Westmead Institute for Medical Research, The University of Sydney, Westmead, NSW 2145, Australia. [6] Experimental Cancer Genetics, Wellcome Trust Sanger Institute, Hinxton, Cambridge CB10 1SA, UK. [7] Royal Prince Alfred Hospital, Camperdown, NSW 2050, Australia. [8] Dermatology Clinic, University Hospital Zürich, University of Zurich, Zurich 8091, Switzerland. [9] Department of Oncology, Clinical Sciences, Lund University, Lund 221 85, Sweden. [10] Departments of Dermatology and Pathology and the Helen Diller Family Comprehensive Cancer Center, University of California, San Francisco (UCSF), San Francisco, CA 94143, USA. [11] Department of Pathology, Fox Chase Cancer Center, Philadelphia, PA 19111, USA. [12] Department of Pathology, University of California, San Francisco, CA 94143, USA. [13] Royal North Shore and Mater Hospitals, Sydney, NSW 2065, Australia. [14] These authors contributed equally: Felicity Newell, Yan Kong, James S. Wilmott. [15] These authors jointly supervised this work: Nicholas K. Hayward, Nicola Waddell, Graham J. Mann, Jun Guo, Richard A. Scolyer. Correspondence and requests for materials should be addressed to R.A.S. (email: richard.scolyer@melanoma.org.au)

Mucosal melanoma arises from melanocytes that reside in a range of internal epithelial tissues of ectodermal origin. The oral cavity, nose and paranasal sinuses, genital tract, and anorectal region are the commonest primary sites for mucosal melanoma. In populations where cutaneous melanoma is common, chiefly those of European background, it accounts for fewer than 1% of all melanomas, but in other populations mucosal melanomas can comprise 25% of all melanomas[1]. Its causes are unknown, although recent genomic studies have supported the concept that solar ultraviolet radiation (UVR) is not the principal carcinogen involved. Studies using exome[2–6] or whole-genome sequencing[2,7] have shown that mucosal melanomas have a much lower burden of point mutations and a greater load of structural chromosomal variants compared to cutaneous melanomas, and that these mutations bear essentially no signatures of UVR or any other known carcinogen. In this they resemble acral melanomas, those of the volar surfaces of hands, feet, fingers, and toes, for which etiology is also unknown.

Mucosal melanomas are challenging to treat. Compared to cutaneous melanoma they are typically detected at a more advanced stage, lack dominant MAP kinase-activating mutations, and respond to immunotherapy less frequently[8,9]. To date, only limited progress has been made in identifying actionable driver mutations for this disease. Alterations to *SF3B1*, *KIT,* and *NF1* are relatively common compared to cutaneous melanomas[7], while mutations to *BRAF* and *NRAS* are less frequent in mucosal melanomas[3,4,10]. Similar to some cutaneous melanomas, *BRAF* fusions occur in mucosal melanoma, although they are rare. Tumors carrying such fusions are somewhat sensitive to anti-MEK targeted therapy, but long-term disease control is rarely achieved[11].

As some of the basic biology of mucosal melanoma remains unclear, limiting both prevention and treatment, here we conduct the largest genomic analysis to date of mucosal melanomas (n = 112) from China, Australia, the United States, and Europe. Using whole-genome sequencing (WGS), we analyze 67 fresh-frozen tumors and validate the key driver genes in whole-exome sequence (WES) data. We identify diverse drivers that indicate the majority of mucosal melanomas are potentially susceptible to CDK4/6 and/or MEK inhibitors.

## Results

**Study sample and approach.** Sixty-seven patients with fresh-frozen tumors were included in the WGS analysis and 45 with FFPE tumors in the validation cohort. Demographic, country of origin, and clinicopathologic details of the 67 patients and their tumors that underwent WGS are presented in Supplementary Data 1. Samples comprised 12 anorectal, 15 female genital, 17 oral, 17 nasal, 2 conjunctival melanomas, and 4 mucosal melanomas of unknown primary site, collected in China (n = 39), Australia (n = 24), Sweden (n = 3), and Switzerland (n = 1). Samples from China were predominantly from upper body sites (31/39) and samples from Australia and Europe were mostly from lower body sites (20/28). Samples were from primary tumors (n = 31) as well as recurrent and metastatic sites (n = 26), with 10 of unknown type. Survival and treatment data are included in Supplementary Data 1; however, due to short follow-up times in the samples from China (all under 2 years, median of 4.4 months), associations of survival with genomic features were not formally analyzed.

Samples underwent WGS to an average read depth of 55× (range 42–100×) in the tumor sample and 31× (range 21–55×) in the matched normal DNA. Ethnicity of patients was determined by using principal component analysis to compare the genotypes of WGS samples with 1000G phase III samples of known

populations (Supplementary Fig. 1). All Chinese samples and one Australian sample (n = 40) clustered with the East Asian ancestry super population. Twenty-four samples clustered with samples of European ancestry and three remaining Australian samples had an ancestry that was not European or East Asian. Genetic ancestry was used for subsequent comparisons between samples in terms of genomic features in this study. Based on genetic ancestry, the age at diagnosis of mucosal melanoma was younger in patients of East Asian ancestry, compared with patients of European ancestry (Mann–Whitney, P = 0.01, mean age 53 vs 63).

The validation cohort derived from FFPE mucosal tumors from the UK and USA comprised 4 anorectal, 21 female urogenital, 3 oral, and 17 nasal tumors, and underwent WES to an average depth of 75× (range 25–106×) in the tumor sample and 88× (range 48–115×) in the matched normal sample (Supplementary Data 2).

**Point mutation burden and mutational signatures.** In keeping with our previous report for this tumor type[7], all mucosal melanomas showed a low single-nucleotide variant (SNV) and insertion/deletion (indel) mutation burden, with an average of 2.7 mutations per Mb (range 0.54–7.1) (Fig. 1a). Tumors from oral, conjunctival, and nasal locations had modestly higher rates than those from the anorectal and urogenital tracts (Mann–Whitney, P = 0.08). Tumor samples that were primary tumors had a significantly lower mutation load than samples from recurrent/metastatic sites (Mann-Whitney, P = 0.033).

When the flanking nucleotide context of these mutations was analyzed, seven single base pair substitution mutational signatures were identified (Fig. 1b, Supplementary Fig. 2a). In contrast to cutaneous melanomas, which typically are dominated (70–95%, Supplementary Fig. 2b) by UVR-related COSMIC signature 7 (ref. [7]), most mucosal tumors showed no or low contribution from signature 7. Interestingly, aside from a conjunctival melanoma, which as would be expected, had a high UVR signature contribution, five other tumors showed >50% contribution (Supplementary Fig. 2b), all of which were from upper mucosal body sites (and of Chinese origin), except one from an unknown primary site (Fig. 1a, c). There were no significant differences in the number of SNVs/indels, structural variants, or percent of the genome affected by copy number aberrations between samples with >50% UVR signature and <50% UVR signature.

The contribution of the ubiquitous age-related signature 1 was more prominent in lower compared with upper body mucosal sites (Mann–Whitney, P < 0.0001) with no significant difference in age between patients with upper or lower body site tumors (Mann–Whitney, P = 0.22) (Fig. 1c). However, lower body site tumors in this study are predominantly from European patients and there was a significant difference in the age at diagnosis between patients of European and Asian ancestry (Mann–Whitney, mean age 63 vs 53, P = 0.01). Three other signatures were found to closely resemble previously identified signatures (as determined by cosine similarity): the ubiquitous signature 5, signature 17, and a signature similar to signature 3, which we termed signature 3-like (Supplementary Fig. 2a) and is a composite of signatures 3, 39, and 40. Signature 3 has been associated with *BRCA1, BRCA2,* and/or *PALB2* mutations[12–14]. However, no pathogenic germline variants or biallelic loss of somatic mutations in *BRCA1*, *BRCA2*, or *PALB2* was identified in the samples with >50% contribution of the signature 3-like signature. Therefore, in mucosal melanoma this signature may be due to contributions from signatures 39 and 40, which have no known etiology. Signature 17, of unknown etiology, was present

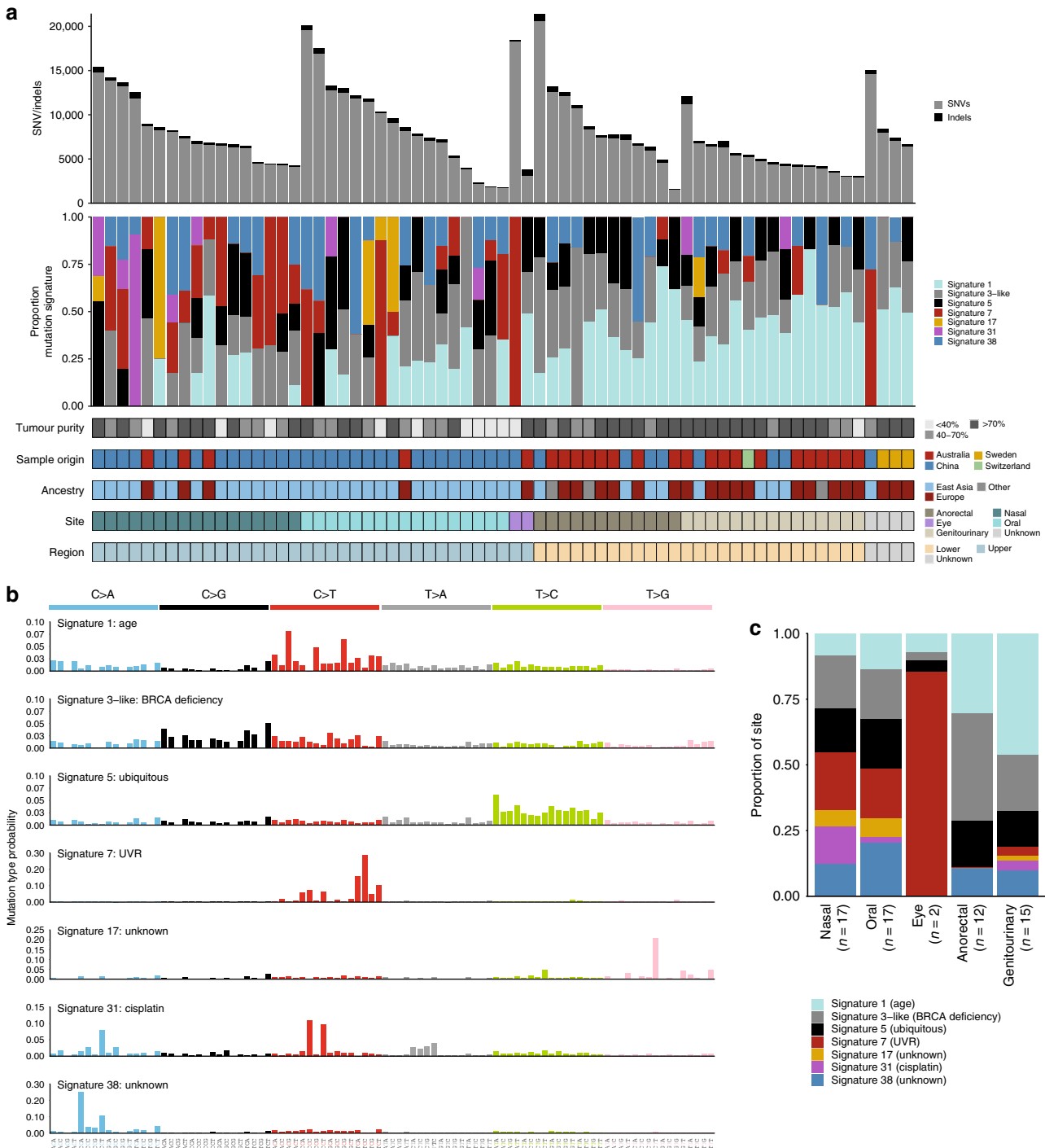

**Fig. 1** Mutational signatures. **a** Mutation burden (top) and proportion of mutational signature (second from top) per sample. The tumor purity, country of origin, site, and region for each sample is shown beneath the plot. **b** Seven mutational signatures were identified in the WGS cohort. For each signature, the mutational type probability for each substitution in a trinucleotide context is shown (total 96 contexts). UVR ultraviolet radiation. **c** Proportion of each signature in tumors from the upper: nasal, oral eye and lower: anorectal and genitourinary body sites. Source data for Fig. 1c are provided as a Source Data file

only in samples ($n = 6$) of Chinese ancestry, and mostly in upper body mucosal sites.

The final two mutational signatures were recently reported signatures by PCAWG Pan-Cancer mutational signature analysis[15]. These signatures are signature 38, which is found almost exclusively in melanomas and is proposed to be caused by an indirect effect of UVR, and signature 31, which was dominant in only one of the mucosal melanomas. Signature 31 (and the closely

related signature 35) has been associated with cisplatinum treatment[16], and the patient with the dominant signature 31 in their melanoma was subsequently confirmed to have received several rounds of cisplatinum chemotherapy prior to their biopsy being removed. Consistent with previous observations, signature 31 displayed strand bias in the opposite direction to the UVR-related signature 7 and had a high proportion of CT > AC dinucleotide substitutions. In contrast, signature 7 exhibited the

typical CC > TT dinucleotide substitutions that are due to UVR (Supplementary Fig. 2c, d).

**Burden and patterns of structural variants**. Tumors had a mean of 260 structural variants (SVs) per tumor (range 21–1300). When classified as rearrangement signatures according to SV spatial distribution and fragment size composition[17], five rearrangement signatures were identified (Fig. 2a) and were compared with previously described rearrangement signatures[17] by cosine similarity. All signatures had a similarity of >0.8 and were identified as RS4 (similarity: 0.99), RS6 (0.97), RS2 (0.98), RS1 (0.85), and RS5 (0.86). RS4 and RS6 are characterized by clustered SV breakpoints, with RS4 having a high number of inter-chromosomal translocations and RS6 having clustered inversions, duplications, and deletions. Unsupervised hierarchical clustering based on the proportions of rearrangement signatures identified two groups of tumors (Fig. 2b). Twenty-eight tumors were characterized by high overall SV counts, clustered breakpoints, a dominance of signatures RS4 and RS6, and a high number of kataegis loci (localized regions of substitution hypermutation) (Group 1). The remaining 39 tumors contained a lower number of SV counts, with dominant unclustered SV signatures RS1, RS2, and RS5 and low numbers of kataegis loci (Group 2) (Fig. 2a). The relative proportion of Group 1 and Group 2 tumors did not differ between upper and lower body location (Fisher's exact, $P = 0.45$), or by sample genetic ancestry (Fisher's exact, $P = 0.24$).

Complex events often targeted chromosomes 5, 11, and 12 and were primarily present in samples in Group 1 (Fig. 2b). A review of chromosomes showing evidence of clustered breakpoints revealed that the localized events had some features of genomic catastrophes. Although there were some tumors that showed patterns similar to chromothripsis (clustered breakpoints, oscillation of copy number, and retention of heterozygosity) and breakage-fusion-bridge (BFB) (loss of telomeric regions and a high number of inversions), most events were too complex to confidently assign to one particular type of mutational mechanism. Other complex focal events involved amplified loci linked together by high numbers of translocations. Focal regions on chromosomes 5p (Supplementary Fig. 3a), 11q (Supplementary Fig. 3b), and 12q (Supplementary Fig. 3c) were particularly targeted, and had a high density of SV breakpoints, and in about 20% of the tumors the regions also contained highly amplified loci. These amplified loci encompassed known melanoma drivers TERT (chr5), CCND1 (chr11), MDM2 (chr12), and CDK4 (chr12)[7,18] (Supplementary Fig. 3d–f), as well as other genes reported to be amplified and/or overexpressed in melanoma[19,20] including SKP2 (chr5) and GAB2 (chr11). Examples of targeted regions in specific samples are shown in Supplementary Fig. 4a–c. Of note, eight samples showed multiple (>5 per sample) translocation events between 5p and 12q (Supplementary Fig. 4d), suggesting that these recurrent events are positively selected. Most of the samples with chromosome 5p–12q translocations were oral mucosal melanomas (7 oral, 1 anorectal), of East Asian ancestry (7 East Asian, 1 European), had amplifications of CDK4 or MDM2 (7/8) on chromosome 12 and TERT or SKP2 (4/8) on chromosome 5 and were, on average, younger at tumor diagnosis when compared with the overall cohort ($P = 0.006$, mean age 42 vs 58). When analyzing oral mucosal samples alone, samples with chr5p–12q translocations also had a lower age at diagnosis (Mann–Whitney, $P = 0.008$, mean age 39 vs 58 any ancestry, $P = 0.01$ samples of East Asian ancestry only), but no difference by gender.

Focal regions of structural rearrangements on chromosomes such as 5p, 12p, and 11p also co-localized with regions of localized hypermutation, termed kataegis[21]. The trinucleotide context of SNVs that fell within kataegis loci displayed

characteristics of mutational signatures 2 and 13 which are associated with APOBEC deamination (Supplementary Fig. 3g). APOBEC signatures were not identified in the mutational signature analysis for the cohort overall (Fig. 1), but this is likely due to the low number of SNVs contained within kataegis loci compared with the overall number of SNVs (0.6%).

**Significantly mutated genes**. Supplementary Data 3 lists all coding SNVs/indels in the mucosal melanomas subjected to WGS. To identify significantly mutated genes (SMGs), we used a consensus approach involving five tools: the Oncodrive suite of tools (OncodriveFM, OncodriveFM, and OncodriveFML), MuSiC2, 20/20+, dNdScv, and MutSigCV. A total of 10 genes were identified as significantly mutated: NRAS (12/67), BRAF (11/67), NF1 (11/67), KIT (10/67), SF3B1 (8/67), TP53 (6/67), SPRED1 (5/67), ATRX (4/67), HLA-A (4/67), and CHD8 (3/67) (Fig. 3a, Supplementary Fig. 5, Supplementary Data 4). The BRAF mutations were diverse (Fig. 3b), but all mutations were in the protein tyrosine kinase domain and most targeted the 594–600 amino acids hotspot region. NRAS mutations were targeted to hotspots on codon 61, which is the dominant hotspot in cutaneous melanoma, and codon 12, a hotspot less commonly mutated in cutaneous melanoma[7,18,22,23] (Fig. 3b). The MAPK pathway-activating mutations were almost completely mutually exclusive, as previously reported[7,18,23]. NRAS mutations were mostly found in samples from recurrent/metastatic sites (two primary, eight recurrent/metastatic, two unknown, Fisher's exact, $P = 0.032$). Interestingly, there was little overlap between tumors with MAPK pathway mutations and SF3B1-mutated tumors, suggesting that the latter mutations may result in MAPK activation. All SF3B1 mutations targeted the 625 codon hotspot (Fig. 3b), and all but one of the SF3B1-mutated tumors originated from anorectal or female genital sites. SF3B1 mutations were also mostly in mucosal melanomas of European ancestry (7/8) and all were from primary tumor samples. BRAF mutations were rare in the nasal cavity, with no codon 600 mutations and only one G-loop mutation identified (G469A). The six tumors with >50% UV signature had no statistically significant difference in driver genes mutations, but lacked mutations in TP53, SPRED1, SF3B1. There were no other relationships evident between SMGs and primary melanoma anatomic site, primary compared with metastatic/recurrent site or patient ancestry.

The SMGs were assessed in a validation FFPE tumor set that underwent WES analysis (Fig. 3c, Supplementary Fig. 5). The HLA-A and CHD8 genes were not mutated in this set and are therefore potential false-positive SMGs, but other genes showed mutations at a similar frequency to the discovery set, with two exceptions: BRAF and SPRED1, each mutated in only 1/45 WES tumors. Once again, SF3B1 mutations were also predominant (5/6) in tumors from anogenital regions.

Since SMG algorithms do not consider promoter mutations, we discuss TERT promoter mutations below in the context of other telomere maintenance genes and mutational mechanisms.

**Other cancer driver gene mutations**. We looked for somatic mutations in other known driver genes, either previously reported for melanoma or in other cancer types. Four WGS samples carried somatic mutations of CTNNB1 (Supplementary Data 3). Two samples were from Australia (an anorectal melanoma with p. L31_I35del and a mucosal melanoma arising in the lacrimal sac with p.S33C) and two were from China (a nasal melanoma with p.T41A and an oral melanoma carrying both p.P44A and p.S45P mutations on the same haplotype). Two samples in the WES validation cohort also had CTNNB1 mutations. All of these mutations occurred at documented hotspots in various cancer

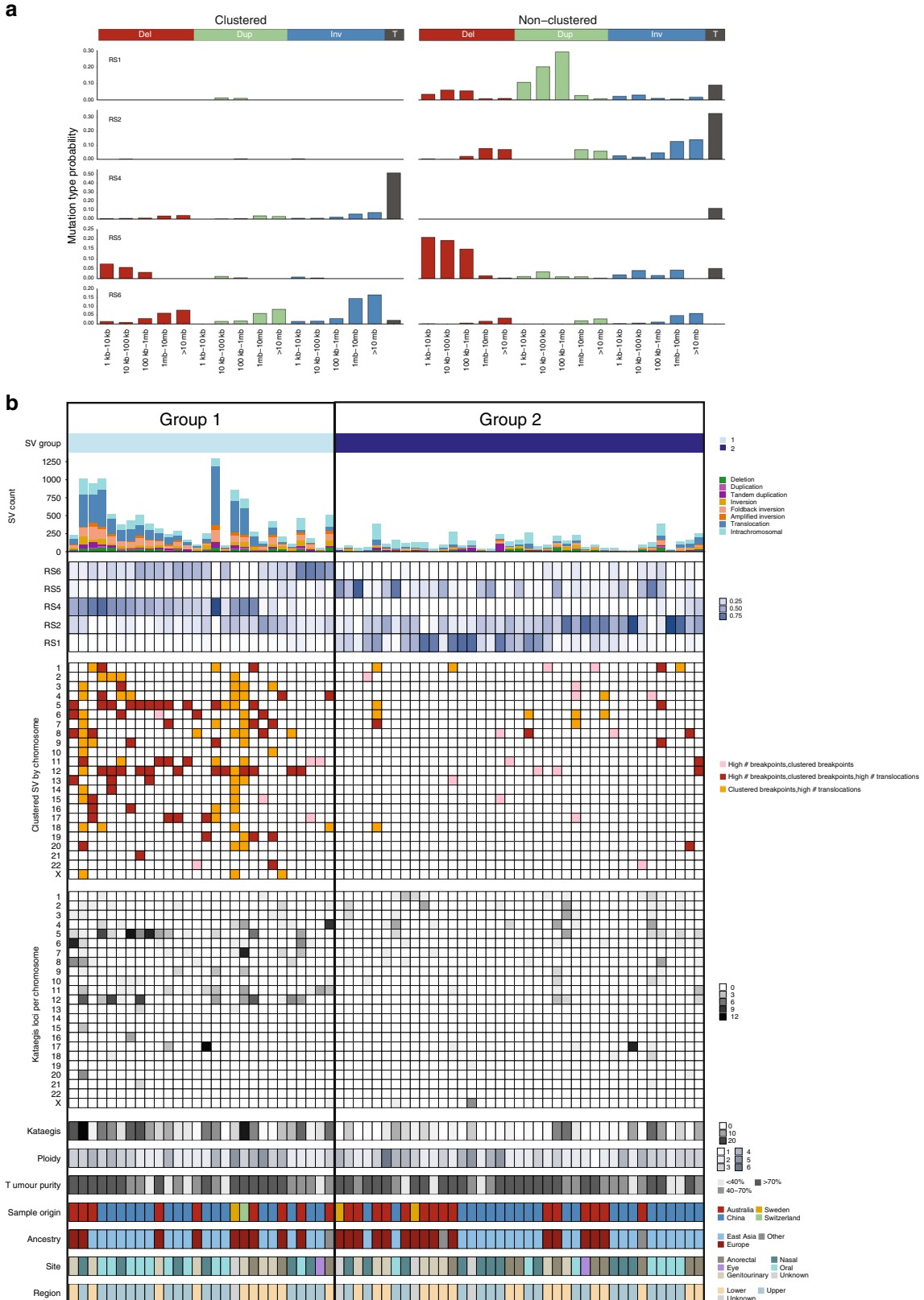

**Fig. 2** Structural variant complexity. **a** Five rearrangement signatures (RS) were identified. Rearrangements were classified into 32 categories based on the rearrangement size, type (Del = deletion, Dup = duplication, Inv = inversion, T = translocation) and whether breakpoints are clustered (left) or non-clustered (right). **b** Two groups of tumors following clustering with ConsensusClusterPlus. Plots from top to bottom are: number and type of SVs; proportion of each rearrangement signature (light blue = lower, dark blue = higher); evidence of localized complexity per chromosome, per sample (light = lower, dark = higher); number of kataegis loci per chromosome, per sample; total number of kataegis loci per sample; tumor ploidy; tumor purity; sample origin, sample ancestry, tumor body site, and body region

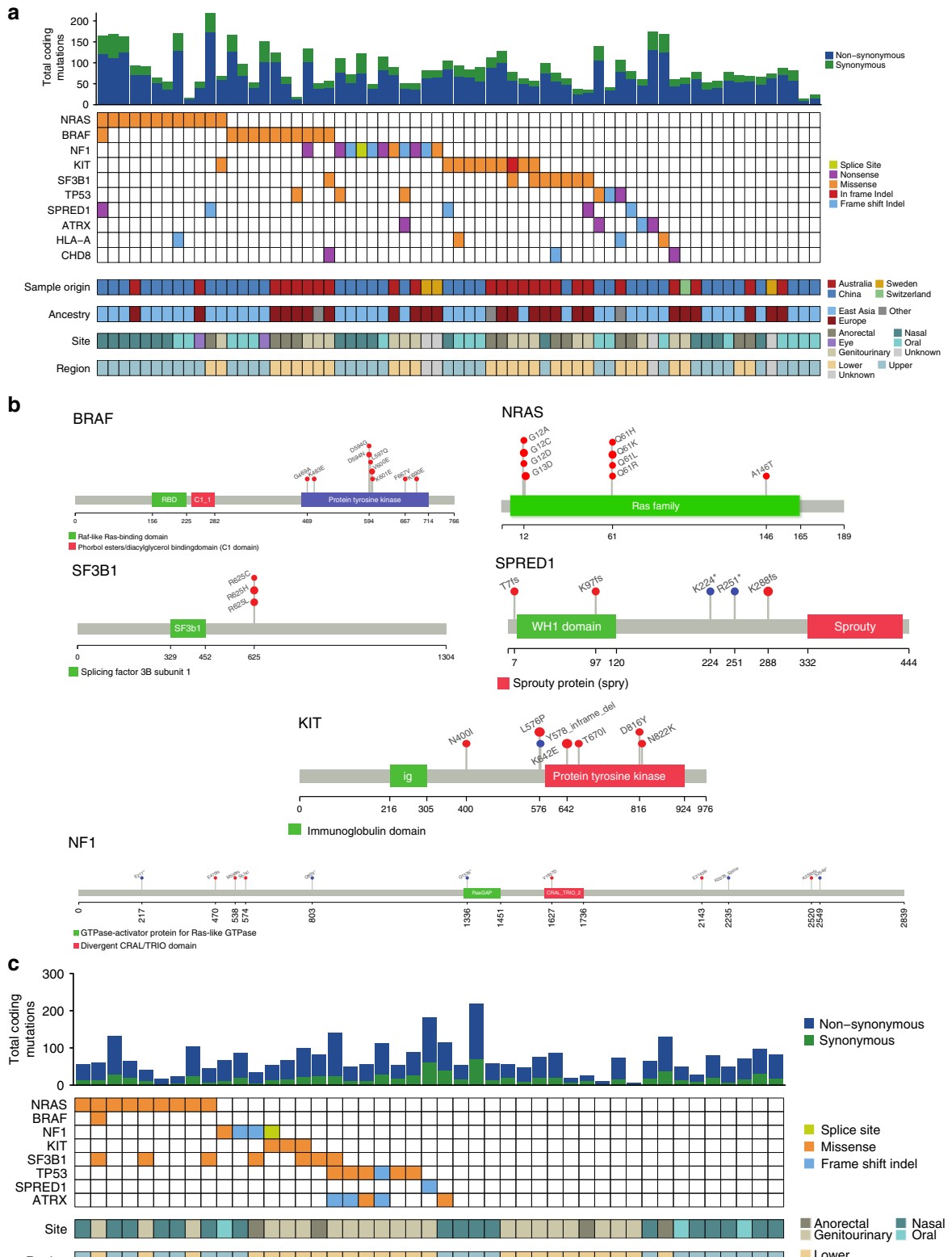

**Fig. 3** Significantly mutated genes affected by SNVs and indels. **a** Number of coding mutations and oncoplot of mutations in 10 significantly mutated genes in the WGS cohort (*n* = 67). If a sample has multiple SNV/indel in a gene, the SNV/indel with the most severe predicted consequence is shown. **b** Positions of *BRAF, NRAS, SF3B1, SPRED*1, *KIT,* and *NF1* somatic mutations in the protein. **c** Number of coding mutations and oncoplot of mutations in eight significantly mutated genes in the WES replication cohort (*n* = 45). If a sample has multiple SNV/indel in a gene, the SNV/indel with the most severe predicted consequence is shown. There were no mutations in CHD8 or HLA-A in the replication cohort

types[24]. Additionally, in single WGS samples, we noted hotspot activating mutations in the oncogenes *MAP2K1*, *GNAQ*, and *KRAS* as well as unambiguous loss-of-function (LoF) mutations in the tumor suppressor genes *CDKN2A* (nonsense), *BAP1* (frameshift), *MEN1* (nonsense), and *NF2* (frameshift).

**Recurrent copy number and structural events.** We then identified genes with significant copy number variations (CNVs) identified by GISTIC, and genes recurrently affected by SV breakpoints that are COSMIC cancer census genes, genes identified by SMG analysis in this study, or previously identified melanoma driver genes (Fig. 4, Supplementary Fig. 6). This analysis revealed strong similarities with melanomas of cutaneous (including acral) sites, with frequent targeting of *TERT*, *CCND1*, *KIT*, *MITF*, *CCND1*, *MDM2*, *CDK4*, and *NOTCH2* by amplification (Fig. 4c, d), and copy loss of *NF1*, *PTEN*, *CDKN2A*, *ATM*, and *ARID1B* (Fig. 4c, e). *CDK4* and *MDM2* were frequently co-amplified on chromosome 12, often along with *TERT*, on chromosome 5, via linking translocations. We observed homozygous deletion of *SPRED1* in two tumors (Fig. 4e). The only gene where CNV abberations was associated with ancestry was *NOTCH2* amplifications, with 4/6 aberrations being in European tumors.

SV breakpoints were also observed in *SPRED1* and breakpoints commonly occurred in melanoma genes *CDKN2A*, *NF1*, and *PTEN*. The SV resulted in 391 predicted fusion events affecting 657 genes (Supplementary Data 5); however, none involved gene pairs that occurred in more than one sample. Two genes (*FAM19A2* and *TENM4*) were involved in four fusion pairs, and four genes (*SHANK2*, *KHDRBS2*, *PTPRR*, and *RGL1*) were involved in three fusion pairs. Only FAM19A2 has the same protein region retained in two samples, but these also have multiple predicted LoF SVs in the gene. Given *FAM19A2* is in the same chromosomal region as *MDM2/CDK4*, these may be passenger events that are not functionally relevant. Aside from a *GRM3* fusion in a mucosal melanoma that we previously reported[7], we observed single gene fusion events affecting *BRAF*, *PAK1*, and *DGKB*, consistent with our observations in cutaneous melanoma[7,25].

**TERT, ATRX, and telomere length.** When relative telomere length (TL) in tumor was compared with TL in matched normal, 56% of mucosal melanomas exhibited TL shortening. We related tumor TL to primary tumor site and mutations in key genes affecting telomere biology[26] (Fig. 5). Putatively activating *TERT* mutations were observed in one-third of tumors (6/67 promoter SNV, 15/67 by high level copy number gain including 1/67 with both promoter SNV and copy number high level gain) and were generally associated with shorter TL (Mann–Whitney, $P = 0.0079$) (Fig. 5, Supplementary Fig. 7a). In contrast, *ATRX* mutations were less frequent (4/67 inactivating SNVs and 3/67 SVs; one with both SNV and SV) and were associated with increased TL ($P = 0.0025$; Supplementary Fig. 7b). There was a trend for *TP53* SNV/indel mutated samples to display telomere elongation ($P = 0.12$; Supplementary Fig. 7c) and when copy number loss (copy number 1) and copy neutral LOH were also considered there was a modest association with telomere elongation (Mann–Whitney, $P = 0.0495$; Supplementary Fig. 7d), as previously reported[26]. There were no associations with other previously reported TL-associated genes *DAXX*, *RB1*, *VHL*, *PBRM1*, or *NRAS* (Supplementary Fig. 7e). Activating *TERT* mutations and loss of function *ATRX* mutations were mutually exclusive, as has also been observed in gliomas[27]. For 4/6 *ATRX*-mutated samples, *TP53* SNV mutations also occurred. In addition, in the WES cohort, all three LoF *ATRX* mutations also co-occurred with *TP53* mutations. When TL was assessed against

putative mucosal cancer drivers identified in this study, a strong correlation with *KIT* activating mutations and telomere shortening was observed ($P = 0.0018$, Supplementary Fig 7f). Lower body site ($P = 0.0022$) (Supplementary Fig. 7g, h) and tumors from patients of European ancestry (Mann–Whitney, $P = 0.013$) were also associated with reduced TL.

**Candidate driver events summary and clinical implications.** Mucosal melanomas carried an average of four (range 0–15) established and putative driver gene aberrations (of all types) (Fig. 6a, Supplementary Data 6). Overall, all but one tumor (66/67) had at least one well-established driver gene mutation (Fig. 6a), i.e. MAPK pathway (*NF1*, *NRAS*, *KIT*, *BRAF*), *SF3B1*, *TP53* and *MDM2*, *SPRED1*, *TERT* and *ATRX*, *CDK4* and *CCND1* (Fig. 6b). When comparing the profile of mutations of mucosal melanomas from East Asian and European ancestry, there were no differences in the presence of driver mutations in any particular genes, with the exception of *SF3B1* which was predominant in patients of European ancestry (Fisher's exact, $P = 0.011$). When comparing any aberration with respect to primary and recurrent/metastatic sample types, *SF3B1* (Fisher's exact, $P = 0.0057$) and *SPRED1* (Fisher's exact, $P = 0.02$) mutations were mostly present in primary samples and *NRAS* aberrations were predominantly in recurrent/metastatic tumors.

Actionable mutations were identified using the Cancer Genome Interpreter website to annotate mutations in the sample that are predicted to confer response (Fig. 6c) or resistance to a targeted agent (Fig. 6d). The inhibitors with the highest levels of evidence included three patients with BRAF V600E where FDA guidelines indicate use of BRAF and/or MEK inhibitors and seven samples with *KIT* mutations. Notably, a large proportion of the cohort (47/67, 70%) harbored mutations potentially responsive to CDK4/6 inhibitors, although the evidence for such treatments is in the lower confidence categories of case reports and early trials. The driver mutations identified as responsive to CDK4/6 inhibitors include *CDK4* and *CCND1* amplifications, as well as *CDKN2A* deletions. In addition, although not identified as a significantly mutated driver, a number of samples also had *CDK6* amplifications (9/67, 13%) that indicates potential sensitivity to CDK4/6 inhibitors. The analysis also identified a *MITF* amplification, *NF1* mutation, and a mutation in *MAP2K1* (encoding MEK1) that may confer resistance to MEK/MAPK inhibitors.

## Discussion

This is the largest analysis of mucosal melanoma performed to date that uses high-coverage WGS to compare melanomas from different body sites and from European and Asian populations. We show different mutational signatures (based on UVR-related and endogenous mutagenic processes) occur in mucosal melanomas arising in facial sites compared to those arising in lower body sites and signatures 7 and 17 occur more often in patients of East Asian ancestry. Additionally, the frequency of specific driver mutations varies with primary melanoma site. For example, *SF3B1* hotspot mutations are common in anorectal and vulvo-vaginal melanomas but are rare in mucosal melanomas from other sites[3,28] and *BRAF* mutations are less common in the nasal cavity. Conversely, while there are two distinct groups of tumors based on the number and distribution of chromosomal SVs, these do not relate to primary melanoma site or genetic ancestry. Together, our results demonstrate that mucosal melanomas show considerable heterogeneity based on the underlying mutagenic processes and body site-specific driver mutations and that genetic ancestry or geographic location may also be factors associated with this.

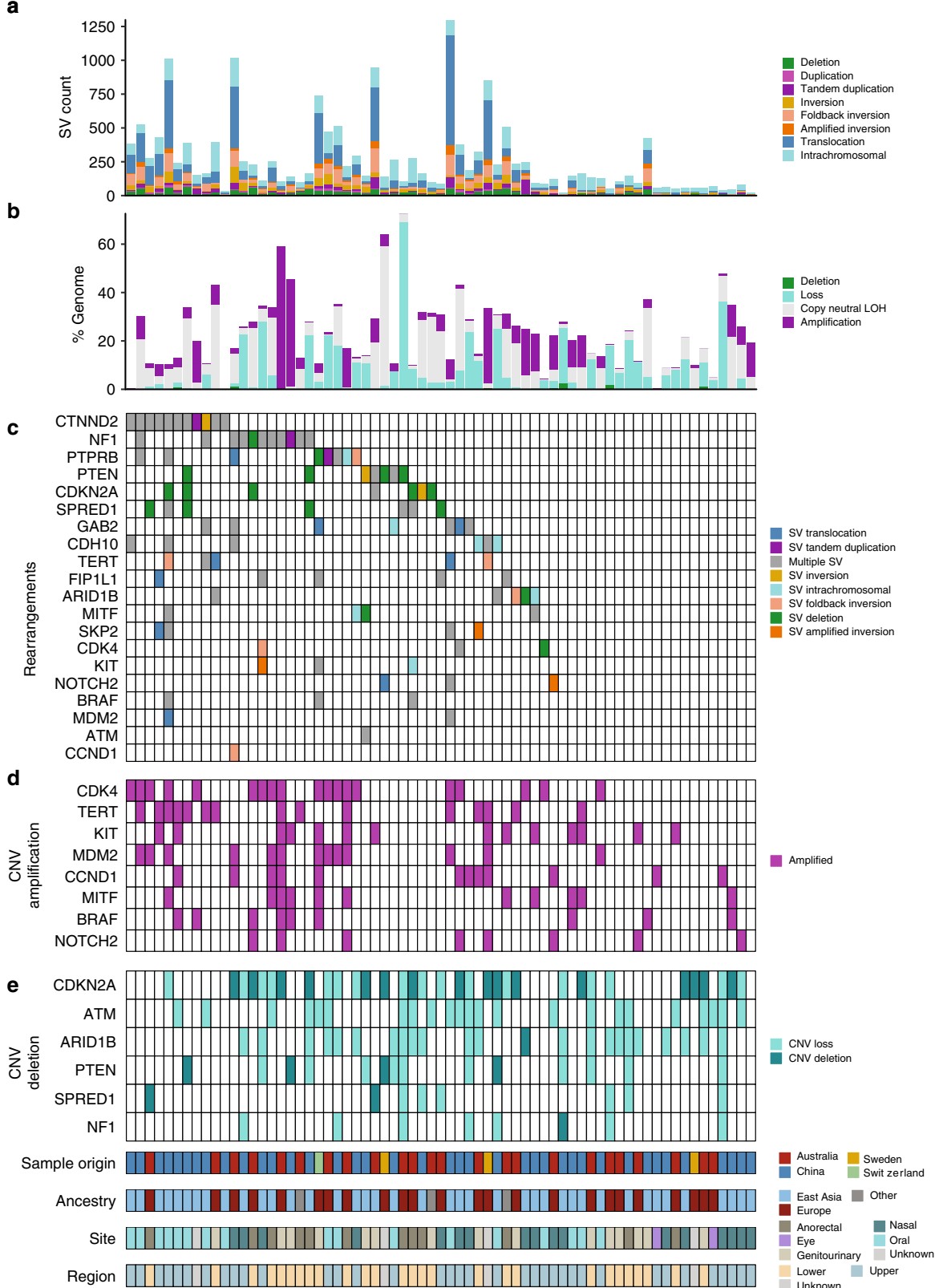

**Fig. 4** Recurrent genes affected by copy number and rearrangement breakpoints. **a** Number and type of SV events. **b** Percent of the genome affected by copy number deletions (CN0), loss (CN1), copy neutral LOH, amplification (CN ≥ 6). **c** Rearrangement breakpoints in genes identified as recurrently affected by SNVs or CNVs are previously identified melanoma driver genes or are COSMIC cancer genes with rearrangements breakpoints that occur in greater than four samples. **d** Copy number amplifications per sample. **e** Copy number loss and deletions per sample

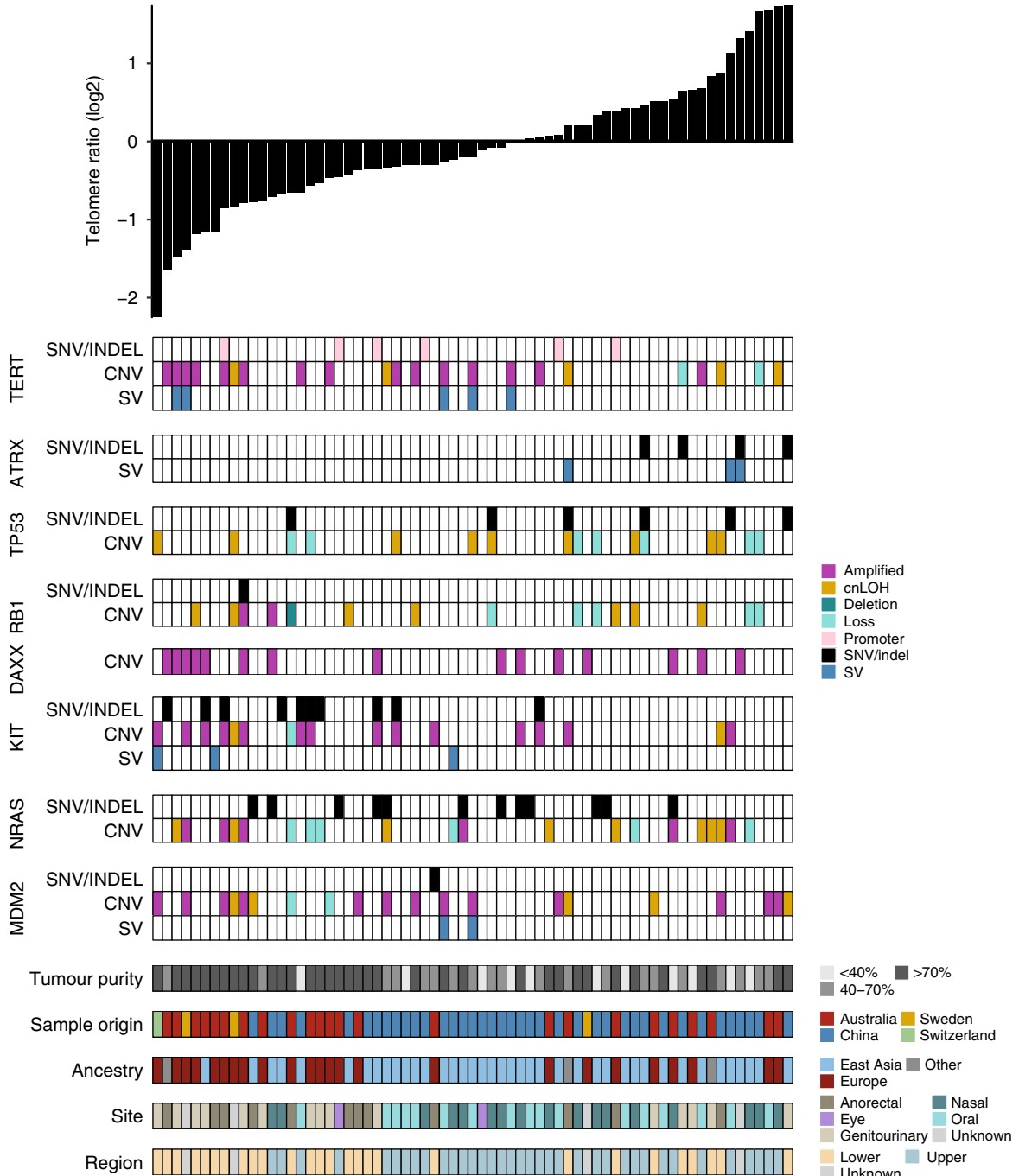

**Fig. 5** Associations between somatic mutations and relative telomere length. Plot of relative telomere length (log2 telomere ratio) per sample is shown at the top and below are plots showing SNV/indels, SV breakpoints, and copy number amplifications (CN ≥ 6, magenta), loss (CN1, light turquoise), deletion (CN0, dark turquoise), and copy neutral LOH (cnLOH) in telomere-associated genes or melanoma-associated genes

The large number of samples from a range of different primary tumor sites enabled us to perform the first detailed characterization of the mutagenic processes that underlie mucosal melanoma. Interestingly, in contrast to prior reports, mutations resulting from UVR exposure were identified in some melanomas from facial mucosal sites while confirmed to be absent from lower body mucosal sites. The presence of a UVR-related signature in a conjunctival melanoma was expected, given the direct exposure of this site to solar UVR; however, our data show that some mucosal melanomas in the nasal cavity, nasal sinuses, and oral cavity have also arisen under a degree of UVR exposure. Nevertheless, when

compared with cutaneous melanomas[7], the UVR-related mutation load of sinonasal and oral mucosal melanomas is low. However, the evidence of UVR-related mutagenesis in mucosal melanomas of the facial area implicates reflected or attenuated UVR exposure as a carcinogen even in these relatively sun-shielded sites. The fact that UVR signatures occur predominantly in samples of East Asian ancestry may be because mostly upper body site tumors in this study were from Chinese patients, or may reflect other geographic factors. Likewise, that signature 17 occurred only in patients of East Asian ancestry also indicates geographically specific environmental or genetic factors may play

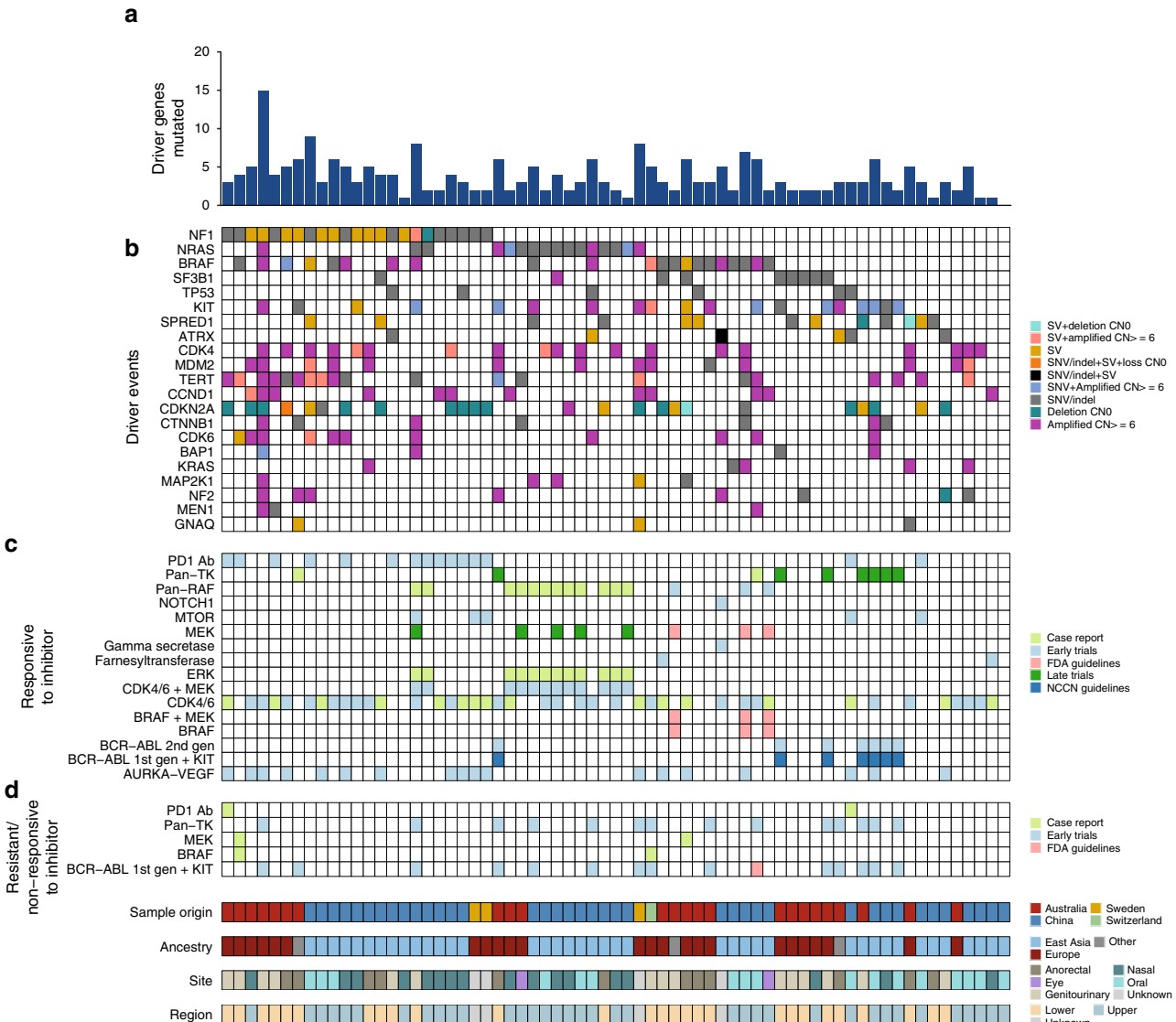

**Fig. 6** Driver summary and actionable mutations. **a** Number of mutations (SNV, indel, CNV, SV) in mucosal melanoma driver genes (defined as SMGs and known oncogenic activating mutations or LoF mutations in tumor suppressor genes in other cancer types) identified in this study. **b** Mutations per sample in driver genes. **c** Samples that have mutations (SNVs, indels, copy number amplification or homozygous deletion or SV fusion gene) that are predicted to be responsive to inhibitors by Cancer Genome Interpreter. Each actionable mutation is colored by evidence level: case report, early trials, late trials, NCNN (National Comprehensive Cancer Network) guidelines, FDA (Food and Drug administration) guidelines. **d** Samples that have mutations (SNVs, indels, copy number amplification or homozygous deletion or SV fusion gene) that are predicted to be resistant or non-responsive to inhibitors by Cancer Genome Interpreter. Each actionable mutation is colored by evidence level: case report, early trials, late trials, NCNN guidelines, FDA guidelines

a role in the development of mucosal melanomas. A larger cohort of samples from each ethnicity would be required to fully elucidate these factors.

Nevertheless, most of the mutation burden in mucosal melanoma cannot be assigned to known carcinogens. The dominant, non-UVR-related mutagenic processes included the clock-like (age-related) mutational processes of signature 1 (spontaneous deamination of 5-methylcytosine), an endogenous process that is active in the vast majority of cancers[15]. Significantly higher proportions of signature 1-related mutations were present in the mucosal melanomas of lower than upper body sites. The process of spontaneous deamination of 5-methylcytosine has been shown to correlate with mitotic divisions and age of cancer diagnosis[15]. While there was no significant difference in age of diagnosis between the patients with upper and lower body mucosal melanomas, patients of East Asian ancestry, which made up the

greater proportion of upper body site tumors were, on average, older than patients of East Asian ancestry. Therefore, further work is required to establish if the presence of more signature 1 in the lower body site tumors reflects real site-specific biology or is a result of the ethnicity of the patients whose tumors samples were examined in our study.

Our multi-tool analysis of SMGs identified *SPRED1* as a driver in mucosal melanoma, which was mutated in tumors from a variety of anatomical sites. *SPRED1* is a tumor suppressor that acts by transporting NF1 to the plasma membrane where it inhibits RAS-GTP signaling. Recently, Ablain et al.[29] reported *SPRED1* loss as a driver of mucosal melanoma. Loss of *SPRED1* function results in increased MAPK pathway signaling[30]. However, while *NF1* and *SPRED1* mutations typically co-occur in cutaneous melanoma[7,31], in our study of mucosal melanomas, most *SPRED1* aberrations identified in WGS samples (11/13)

occurred in *NF1* wild-type samples, with only two SVs in *SPRED1* co-occurring with *NF1* mutations.

As previously reported[7,10,31], we found a substantial proportion of mucosal melanomas had aberrations of the MAPK pathway, including frequent mutations in *NRAS*, *BRAF*, *NF1*, and *KIT*. Mutations were also identified in several other known cancer driver genes, including *CTNNB1*, implicating the previously unappreciated role of WNT signaling in the genesis of mucosal melanomas. Since activation of the Wnt/β-catenin pathway has also been demonstrated to cause T cell exclusion[32], *CTNNB1* mutations may also contribute to the relatively poorer responses to immunotherapy observed in some advanced stage mucosal melanoma patients[33,34] Hotspot mutations were also identified in the oncogenes *MAP2K1* and *KRAS*, which together with a *BRAF* fusion, further highlight the reliance on MAPK pathway activation in this tumor type.

Our study provides important insights into the complexity and diversity of the SV load in mucosal melanomas. Many of the SV events targeted well-known melanoma driver genes (*CDKN2A*, *NF1*, *PTEN*, and *TERT*), exemplifying their role in mucosal melanoma pathogenesis. We found that there were two distinct subgroups of mucosal melanoma primarily distinguished by their degree of localized complex chromosomal rearrangement. However, these clusters did not co-segregate with upper and lower primary body sites or genetic ancestry suggesting they arise via alternate common pathways of genome instability some time after initiation. Regions of localized complexity could be the result of genomic catastrophes such as chromothripsis. We identified tumors that showed patterns similar to chromothripsis, as well as BFB. In addition, we identified a number of tumors with regions of amplified loci on multiple chromosomes linked by high numbers of translocations. We have previously observed similar patterns in ovarian cancer[13], and these patterns are characteristic of double minute chromosomes[35,36] or neochromosomes[37]. A number of mucosal melanomas in our cohort showed a pattern of translocations involving 12p amplifications which is similar to that of neochromosomes in liposarcomas[37].

Localized structural rearrangement events targeted a number of regions, including 5p, 11p, and 12p, with a common consequence of these complex events being amplification of oncogenes such as *TERT*, *MDM2*, *CCND1*, *CDK4*. These regions have also recently been identified as significantly recurrent breakpoint regions (SRB) with gene amplifications, and result from positive selection rather than by genome fragility[38]. Therefore, amplification of oncogenes or other cancer genes conferring a growth advantage may be an important mechanism driving mucosal melanoma.

All but one of the eight tumors in this study with translocations between 5p and 12p, usually resulting in amplifications of *MDM2/CDK4* and *TERT*, were oral mucosal melanomas and of East Asian ancestry. Interestingly, oral mucosal melanomas carrying such translocations occurred at significantly younger age than those without translocations. This finding suggests molecular diversity in predominant driver genes between mucosal melanomas from different body sites. This notion is supported by Lyu et al.[4], who recently reported amplifications of 12q14 and 5p15 in ~50% of oral mucosal melanomas, and 6/19 tumors had amplification of both regions.

One potential consequence of genomic instability in mucosal melanomas is an effect on telomere maintenance and length. A key event in cellular immortality is the presence of a telomere maintenance mechanism to escape cellular crisis, potentially occurring through activation of telomerase (*TERT*) or through alternative lengthening of telomeres (ALT), resulting from *ATRX* inactivation. We identified mutually exclusive loss of function *ATRX* mutations and putatively activating *TERT*

promoter mutations and amplifications, suggesting that both maintenance mechanisms are important in distinct subsets of mucosal melanoma. The association of *KIT* with TL in this cohort has not previously been reported, and further work is required to identify the mechanism behind this association.

The finding of *SPRED1* as a driver increases the proportion of mucosal melanoma samples with a known driver event in the MAPK pathway (*NF1*, *NRAS*, *BRAF*, *SF3B1*, *TP53*, *KIT*, and *SPRED1*) to 92% of the studied patients. MEK/MAPK inhibition may thus be suitable for a large proportion of mucosal melanoma patients and warrants evaluation in clinical trials. Driver events affecting CDK4 in a large proportion of our cases points to the potential therapeutic option of using CDK4 inhibitors to treat this tumor type. Indeed, Zhou et al.[39] have very recently shown that such inhibitors are effective in treating patient derived xenografts of mucosal melanomas carrying CDK4 aberrations, thus providing experimental support for this notion. Immunotherapies are standard of care for metastatic cutaneous melanoma patients; however, mucosal melanoma patients treated with PD-1 antibodies respond half as often as cutaneous melanoma patients[8]. It is likely that the low mutation burden in mucosal melanoma is an important factor in limiting responses to immunotherapies in mucosal melanoma patients. Our data also suggest that activation of the β-catenin pathway (through mutations in *CTNNB1*) may be a contributing factor[32]. Improved understanding of the factors regulating immunogenicity in mucosal melanomas is needed to improve outcomes for mucosal melanoma patients, along with comprehensive molecular analysis of both point and structural mutations to reveal all therapeutic opportunities in this difficult and diverse class of tumors.

## Methods

**Human melanoma samples.** All fresh frozen and FFPE samples were obtained in a method that was compliant with the relevant ethical regulations for work with human participants. The fresh-frozen tissue and matching blood samples ($n = 67$) analyzed in the current study were obtained from the biospecimen bank of Melanoma Institute Australia (MIA) ($n = 24$), Peking University Cancer Hospital & Institute, Beijing, China ($n = 39$), the Department of Surgery, Skåne University Hospital, Sweden ($n = 3$), and the Biobank of the University Research Priority Program in translational cancer research (URPP) at the University of Zurich Hospital, Switzerland ($n = 1$). All tissues and bloods form part of prospective collections of fresh-frozen samples accrued with written informed patient consent. The study was approved by the Sydney Local Health District RPAH zone ethics committee (Protocol No. X15-0454—previously X11-0289 and HREC/11/RPAH/444; Protocol No X17-0312—previously X11-0023 and HREC/11/RPAH/32; and Protocol No X15-0311—previously X10-0300 and HREC/10/RPAH/530) and the relevant cases were approved by the institutional ethics committees of Peking University Cancer Hospital & Institute (Beijing Cancer Hospital), Skåne University Hospital and University of Zurich. Fresh surgical specimens were macro-dissected and tumor tissues were procured (with as little contaminating normal tissue as possible) and snap frozen in liquid nitrogen within 1 h of surgery. All samples were pathologically assessed prior to inclusion into the study, with samples requiring greater than 80% tumor content and less than 30% necrosis to be included. All samples were independently reviewed by expert melanoma pathologists R.A.S., P.M.F. and T.J.D. to confirm the presence of melanoma and fulfillment of the above criteria as previously described[40]. All cases were reviewed centrally by the study pathologists (R.A.S., P.M.F., and Z.L.) to confirm the origin of each tumor from wet mucosa, rather than the adjacent oral, nasal, anal, or genital skin. Samples requiring tumor enrichment underwent macrodissection or frozen tissue coring (Cryoxtract, Woburn, MA, USA) using a marked H&E slide as a reference. Mucosal melanomas were defined as occurring in the mucosal membranes lining oral, respiratory, gastrointestinal, and urogenital tracts. The H&E slides of the primary melanomas were reviewed in all cases and any tumor that had arisen in the junction of mucosal and cutaneous skin was excluded.

A validation set of human mucosal melanoma cases ($n = 45$) (defined as occurring in the mucosal membranes lining oral, respiratory, gastrointestinal, and urogenital tracts) were obtained as FFPE tissue from three clinical centers: University of Michigan, University of Edinburgh and University of California, San Francisco as described by Wong et al.[41]. All FFPE exome cases were ethically approved by local Institutional Review Boards at the University of Michigan, University of Edinburgh and University of California, San Francisco and by the Sanger Institute's human materials and data management committee. All samples

were reviewed by specialist dermatopathologists prior to inclusion into the study. Cores were taken from samples using a marked H&E slide as a reference.

**DNA extractions**. Fresh-frozen tumor DNA was extracted using DNeasy Blood and Tissue Kits (69506, Qiagen Ltd), according to the manufacturer's instructions. Blood DNA was extracted from whole blood using Flexigene DNA Kits (51206, Qiagen Ltd). All samples were quantified using a NanoDrop (ND1000; Thermoscientific) and Qubit® dsDNA HS Assay (Q32851; Life Technologies) and DNA size and quality were tested using gel electrophoresis. Samples with a concentration of less than 50 ng/μl, or absence of a high molecular weight band in electrophoresis gels, were excluded from further analyses.

DNA from tumor samples obtained as FFPE tissue cores was extracted with QIAamp FFPE Tissue kits (Qiagen), according to the manufacturer's instructions and quantified using the Qubit® dsDNA HS Assay (Q32851; Life Technologies)[41].

**WGS, processing, and quality control**. Sixty-seven patients underwent whole-genome paired-end sequencing on a HiSeq2000 or HiSeq X Ten (Illumina, San Diego, CA, USA) at The Kinghorn Cancer Centre, Garvan Institute of Medical Research (Sydney, Australia), Macrogen (Geumcheon-gu, Seoul, South Korea), or Novogene Bioinformatics Technology Co. Ltd, China. Library construction was performed using TruSeq DNA Sample Preparation kits (Illumina) as per the manufacturer's instructions. All downstream processing, including sequence alignment and variant calling, was carried out at QIMR Berghofer (Brisbane, Australia) using the same analysis pipeline for all whole-genome samples. Sequence data were adapter trimmed using Cutadapt (ref.[42]; version 1.9) and aligned to the GRCh37 assembly using BWA-MEM (version 0.7.12) and SAMtools (version 1.1). Duplicate reads were marked with Picard MarkDuplicates (https://broadinstitute.github.io/picard (version 1.129); http://picard.sourceforge.net). Assessment of the sequencing and alignment quality of the sample was carried out using qProfiler (version 1) and estimation of coverage using qCoverage (version 0.7pre; http://sourceforge.net/projects/adamajava). All tumors had a minimum tumor purity of 15%. Tumor purity was assessed using ascatNGS. Where ascatNGS was unreliable (following manual review), mean variant allele frequency was used (Supplementary Data 1).

**Whole-exome sequencing, processing, and quality control**. The 45 FFPE-derived tumor samples that formed the validation cohort underwent whole-exome sequencing at the Wellcome Trust Sanger Institute (Hinxton, United Kingdom)[41]. Sequencing libraries were generated using Agilent SureSelect All Exon V5 kits and sequenced on the HiSeq2500 platform (Illumina, San Diego, CA, USA) to generate 75 bp reads. Sequence data were aligned to the human genome assembly GRCh38 using BWA-MEM (version 0.7.12) and Biobambam (version 2.0.18) was used to mark PCR duplicates. BAMs were merged to sample level using SAMtools[43]. To ensure that tumor and matched normal samples had the best reciprocal match, SAMtools mpileup, followed by BCFtools "gtcheck" was run to detect potential sample swaps and contamination. To assess sequencing and alignment quality, the SAMtools "stats" utility was used to determine the coverage and PCR duplicate rate in bait regions plus 100 bp flank on either side. Quality filtered BAM files were generated for downstream analysis by removing PCR duplicates, reads that failed Illumina chastity filtering supplementary and secondary read alignments. In addition, reads were removed where two reads had the same start position and one read had a mapping quality of zero, and mapped randomly to a repetitive site, an artifact frequently observed in libraries from FFPE samples. Assessment of sample swaps and contamination using BCFtools and coverage metrics was performed again on the quality filtered BAM files.

**Whole-genome somatic SNV/indel analysis**. Somatic SNV and indels were detected using an established pipeline[7] where a dual calling strategy was used to detect SNV, with the consensus of two different tools being used for downstream analysis: qSNP (version 2.0)[44] and GATK HaplotypeCaller (version 3.3-0)[45]. Detection of indels (1–50 bp) was carried out using GATK. Variant annotation for gene consequence was performed using SnpEff.

**Whole-exome somatic SNV/indel analysis**. Somatic SNV were called using MuTect (v1.1.7)[46]. MuTect was run with some non-default parameters to deal with contamination of normal samples with tumor DNA and with quality issues seen in FFPE samples. Parameters used included: allowing the normal sample to have <5% of reads with an alternate allele and <4 reads total with the alternate allele; allowing for some clustering of alternate alleles at a consistent distance from the start/end of the read by decreasing the requirement for exclusion based on the fraction of read that is clipped from 0.3 to 0.25; and increasing the minimum base quality score required in order for a base to be considered. The base quality score threshold was changed because an artifact (from either PCR or sequencing) was identified where both low-quality reference bases and mismatched bases were consistently present downstream of polynucleotide runs. MuTect variants were then filtered using tiered BCFtools filtering based on read depth and variant allele fraction (VAF). SNVs were excluded if they had either: a

depth of less than 10; VAF < 0.20 with depth < 20; VAF < 0.05; VAF > 0.15 and number of supporting reads <5; VAF < 0.30 and normal contained 3 or more alt reads; or normal had 2 (or more) alt reads and the tumor VAF was <0.20. Additional filters were applied to MuTect variants, including removing variants that are common (>0.01) in gnomAD[47] and using SAMtools mpileup to check for alternate alleles in unrelated normal samples. Variants for which there were three or more alternate alleles in at least two unrelated normal samples were identified as falsely somatic and excluded.

As MuTect does not identify multi-nucleotide variants (MNVs), MAC v1.2 software (Multi-nucleotide Variation Annotation Corrector; https://github.com/hubentu/MAC) was applied to the variants identified by MuTect. MAC takes a list of SNVs, checks adjacent and nearby SNVs, and phases them. In-house scripts were used to take output from MAC, rerun variant effect prediction on MNVs, and fix VCF entries. Somatic indels were called using Strelka (v1.0.15)[48]. Indels were filtered to remove variants that were common (>0.01) in gnomAD. Mutation consequences for SNVs and Indels were predicted using Ensembl Variant Effect Predictor (VEP)[49] and gene models from Ensembl release v89.

**Determination of sample genetic ancestry**. To determine genetic ancestry, mucosal sample genotypes were compared with the genotypes of populations examined in the 1000 genomes (1000G) project[50]. Phase III 1000 genomes genotypes in plink format were downloaded from the Plink2.0 website (https://www.cog-genomics.org/plink/2.0/resources#1kg_phase3). For the 67 mucosal WGS samples, pileups were performed at 1000G SNP positions and a VCF file generated. A genotype was assigned using the following criteria: (i) homozygous reference genotype (0/0) if the variant allele fraction (VAF) was ≤0.2; (ii) heterozygous (0/1) if the VAF was >0.2 and <0.8; (iii) homozygous (1/1) for the alt allele if the VAF was ≥0.8. Positions for which the coverage was less than 10 for any sample were excluded. The VCF file was converted to a plink file and an ancestry check was performed as per the method outlined in the plinkQC R package "Ancestry check" vignette. All subsequent analysis was performed using plink version 1.90b6.8. Genotypes of mucosal samples were pruned for variants in linkage disequilibrium with an $r^2$ greater than 0.2 in a 50 kb window and ranges with high-LD were also removed. The genotypes of 2504 reference (ie 1000G) samples and 67 mucosal samples were merged after correcting for chromosome or position mismatches and updating allele flips. Variants with a missing rate of greater than 0.1 or minor allele frequency of less than 0.05 were removed. Principal component analysis was performed using 39,156 variants and principal components 1 and 2 were then plotted using a plotting function in plinkQC (Supplementary Fig. 1). Samples that did not cluster with European or East Asian (i.e. Chinese populations) were assigned as Other.

**Mutational signatures**. The non-negative matrix factorization (NMF) method described by Alexandrov et al.[12] was used to detect mutational signatures in WGS samples. To determine the contribution of each signature to a sample, we used the quadratic programming approach available in the R package, *SignatureEstimation*[51] to assign the mutations in each sample to the signatures identified by NMF. To prevent over-fitting, signatures that contributed less than 10% for a sample were removed and mutations were reassigned to the signatures that remained. Signatures that were obtained were compared with the profiles of signatures (SBS1–SBS60) found in the recent analysis of more than 4000 whole cancer genomes by the Pan Cancer Analysis of Whole-Genomes network (PCAWG)[15]. The presence of transcriptional strand bias in signatures was assessed using the R package 'MutationalPatterns'[52]. Mutational signatures were not analyzed in the FFPE validation cohort due to low mutation number.

**SMG analysis**. We used a consensus approach and multiple tools for discovery of SMGs affected by SNVs/indels. These tools comprise the Oncodrive suite of tools[53,54], MuSiC2 (ref.[55]), 20/20+[56], dNdScv[57], and MutSigCV. Unless otherwise specified all tools were run using default parameters. The Oncodrive suite consisted of three methods: OncodriveFM, OncodriveClust, and OncoDriveFML. OncodriveFML was executed using CADD v1.0 through the web interface at: http://bbglab.irbbarcelona.org/oncodrivefml/home. OncodriveFM and OncodriveCLUST were executed through the web interface at: https://www.intogen.org/. In order to be considered significant for this tools, a q-value of 0.05 for OncodriveFM and OncodriveCLUST and 0.1 for OncodriveFML were used. MuSic2 (v0.2) was run using a region of interest (ROI) file for hg19 (obtained from https://github.com/ding-lab/calc-roi-covg). A gene was considered significant if it had an FDR for the Fisher's combined P value test of <0.05. For MutSigCV and dNdScv, genes were considered significant at a q-value of <0.1. The tool 20/20+ was run for 10,000 iterations, using the pre-trained classifier "2020plus_10k.Rdata". P value QQ-plots were reviewed and confirmed that the predictions were not inflated for false positives (mean absolute log2 fold change (MLFC) was less than 0.3). A gene was significant if the oncogene, tumor suppressor gene, or driver genes q-values were less than 0.05. A consensus list of 10 significant genes was obtained by considering those genes that were significant in at least two different tools (where the Oncosuite methods were

considered to be one tool). Protein lollipop mutation diagrams were generated using the tool "lollipops"[58].

**Structural variant and copy number analysis**. Structural variants were determined using qSV[35]. Structural variant breakpoints and potential consequence of the SV, including potential in-frame gene fusions, was determined by annotation against Ensembl known genes (version 75) using in-house scripts. The presence of fusion events in COSMIC Cancer census genes and in kinase genes was assessed. The list of COSMIC Cancer census genes were downloaded 20 December 2017 (https://cancer.sanger.ac.uk/cosmic). All Tier 1 and 2 genes were considered. Protein kinases were downloaded from Uniprot 20 December 2017 (http://www.uniprot.org/docs/pkinfam.txt). For genes recurrently affected by rearrangement breakage sites, COSMIC cancer genes that were rearranged in at least four patients were included. Genes that were genes identified by SMG analysis in this study, or are previously identified melanoma driver genes, were also included for SV analysis.

Copy number was determined using sequencing data and the tool ascatNGS[59]. Copy number loss (copy number 1), homozygous deletion (copy number 0), and amplifications (copy number ≥ 6) were considered in the analysis. Copy number per gene was determined by annotation against Ensembl known genes (version 75). Significantly mutated copy number regions were assessed using GISTIC2.0. A gene was considered significant if it was in a focal region with a confidence level of 0.95 and a q-value <0.1. A short list of genes was derived by manual curation and filtering for those genes that are considered Cosmic Cancer Census genes.

**Rearrangement signatures and clustering**. We used the same statistical framework using NMF that was used for mutational signature analysis for the identification of rearrangement signatures[12]. SVs were classified into the same categories as has been described and applied to a breast cancer cohort by Nik-Zainal and co-workers[17]. SVs were classified into types of events: deletions, duplications, inversions, and inter-chromosomal translocations. SVs were further characterized by size and whether the breakpoints were clustered or non-clustered. Size categories (for events that were not translocations) were 1–10 kb, 10–100 kb, 100 kb–1 Mb, 1–10 Mb, more than 10 Mb. Clustered SV breakpoints were defined using the BEDTools cluster function. Clustered events were defined using the presence of ≥10 breakpoints in a 1 Mb window, a metric that has previously been applied by Letouze et al.[60]. The presence of clustered breakpoints on a per-chromosome basis[13]: chromosomes that had a highly significant non-random distribution of breakpoints with a stringent threshold of $P < 10^{-5}$ were considered to be clustered. Chromosomes with high numbers of rearrangement events were identified as outliers defined as a breakpoint per megabase rate exceeding 1.5 times the length of the inter-quartile range from the seventy-fifth percentile for each sample with a minimum threshold of 35 breakpoints per chromosome. Chromosomes with at least 10 translocations were defined as having a high number of translocations.

Unsupervised hierarchical clustering of rearrangement signatures was performed using the R package "ConsensusClusterPlus"[61]. The data used for the clustering were the proportion of rearrangements assigned to the five signatures. The proportions for each signature were mean centered before clustering and the following settings were used: pItem = 0.9, pFeature = 0.9, Pearson's distance metric, number of repetitions = 1000.

**Detection of kataegis**. Localized regions of hypermutation, known as kataegis, were identified using previously established metrics[13]: inter-mutational distance was calculated as the number of base pairs between mutations ordered by chromosome and position. Inter-mutational distances were segmented using piecewise constant fitting and putative regions of kataegis were defined as segments containing six or more consecutive mutations with a mean inter-mutation distance of ≤1000 bp.

**Telomere length**. TL was determined using sequencing data and the tool qMotif[7]. qMotif is freely available at http://sourceforge.net/projects/adamajava and counts the number of reads containing the telomeric repeat (TTAAGG). Counts are normalized to the mean genomic coverage of a sample and the relative TL is expressed as the log2 ratio of read counts in the tumor BAM file to the matched normal BAM file read counts.

**Clinically actionable mutations**. The Cancer Genome Interpreter website (www.cancergenomeinterpreter.org)[62] was used to identify clinically actionable mutations. Mutations that were annotated were: coding SNVs/indels, copy number alterations that were amplifications (copy number ≥6) or homozygous deletions (copy number 0), and predicted inter-gene fusions. The output from the Cancer Genome Interpreter (drug_prescriptions.tsv) was filtered as follows: Alterations = complete, Tested tumor = Contains CANCER or CM. An evidence level that was preclinical was removed. References to BRAF inhibitor resistance was removed in samples that did not contain a BRAF mutation.

**Statistical analysis**. Statistical analyses were performed in R, and p-values < 0.05 were considered statistically significant. Two-sided Fisher's exact tests were performed where specified. If not specified otherwise, two-sided Mann–Whitney U statistical tests were carried out. Sample sizes are listed in figures or figure legends. Boxplots show the median and the 25th and 75th percentiles.

**Reporting Summary**. Further information on research design is available in the Nature Research Reporting Summary linked to this article.

## Data availability
Sequence data that support the findings of this study have been desposited in the European Genome-phenome Archive (EGA). The sequencing data for whole-genome sequenced samples (from Australia, Sweden, Switzerland, and China) are available under study accession EGAS00001001552. The raw sequencing data for the UK/USA WES samples are available under study accession EGAS00001001115. The source data underlying Fig. 1c and Supplementary Fig. 5 are provided as a Source Data file.

## Code availability
Tools used in this publication that were developed in-house are available from the SourceForge public code repository under the AdamaJava project (http://sourceforge.net/projects/adamajava/). Updated versions of software are available at https://github.com/AdamaJava.

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

## Acknowledgements

This work was supported by Melanoma Institute Australia (MIA), Bioplatforms Australia, New South Wales Ministry of Health, Cancer Council New South Wales, Program Grants of the National Health and Medical Research Council of Australia (NHMRC), Cancer Institute New South Wales, and by the Australian Cancer Research Foundation. N.W., J.S.W., N.K.H., and R.A.S. are supported by NHMRC Fellowships. K.N. is supported by a Keith Boden Fellowship. Collection of the Australian samples was supported by MIA. The authors gratefully acknowledge the support of colleagues at MIA, Royal Prince Alfred Hospital, NSW Health Pathology, the Westmead Institute for Medical Research. T.J.D. was supported by the Jani Haenke Melanoma Pathology Fellowship. P.M.F. was supported by the Deborah and John McMurtrie MIA Pathology Fellowship. G.V.L. is supported by an NHMRC Practitioner Fellowships Grant and University of Sydney, Sydney Medical School Foundation Grant. B.C.B. is supported by NIH/NCI grant number 1R35CA220481.

## Author contributions

Conceptualization: I.Y., H.W., N.J., B.C.B., D.J.A., N.K.H., N.W., G.J.M., J.G., and R.A.S.; methodology: F.N., Y.K., J.S.W., P.A.J., P.M.F., C.L.C., Z.L., H.B., T.J.D., A.P., K.N., P.M.S., R.V.R., J.V.P., and R.A.S.; Software, F.N., S.H.K., O.H., C.L., S.W., Q.X., and J.V.P.; validation, I.Y., B.C.B., and D.J.A.; formal analysis, F.N., J.S.W., P.A.J., A.P., K.N., L.V.D.W., K.W., P.M., D.J.A., and N.W.; investigation, F.N., Y.K., J.S.W., X.W., N.W., and R.A.S.; resources, Y.K., J.S.W., P.M.F., C.L.C., Z.L., S.H.K., V.T., P.M.S., O.H., S.L., C.L., S.W., Q.X., R.V.R., R.D., M.P.L., G.J., X.W., I.Y., H.W., N.J., B.C.B, G.V.L., A.J.S., K.F.S, J.F.T., R.P.M.S., D.J.A., L.S., J.V.P., G.J.M., J.G., and R.A.S.; data curation, Y.K., J.S.W., P.M.F., S.H.K., H.B., T.J.D., L.V.D.W., K.W., O.H., S.L., C.L., S.W., Q.X, D.J.A., L.S., J.V.P., J.G., and R.A.S.; writing—original draft, F.N., J.S.W., P.A.J., N.K.H., N.W., G.J.M., and R.A.S.; writing—review & editing, F.N., J.S.W., P.A.J., A.P., K.N., L.V.D.W., K.W., G.J., G.V.L., J.F.T., R.P.M.S., D.J.A., N.K.H., N.W., G.J.M., J.G., and R.A.S.; visualization, F.N. and P.A.J.; supervision, J.S.W., D.J.A., J.V.P., N.K.H., N.W., G.J.M., J.G., and R.A.S.; project administration, Y.K, J.S.W., D.J.A., N.K.H., N.W., G.J.M., J.G., and R.A.S.; funding acquisition, N.K.H., G.J.M., J.G., and R.A.S.

## Additional information

**Competing interests:** J.V.P. and N.W. are founders and shareholders of genomiQa Pty Ltd, and members of its Board. B.C.B. is a Consultant to Lilly Inc. D.J.A. is a paid consultant for Microbiotica and receives research funding support from OpenTargets and BMS. None of these relationships involve the work described in this manuscript. R.D. has intermittent, project focused consulting and/or advisory relationships with Novartis, Merck Sharp & Dhome (MSD), Bristol-Myers Squibb (BMS), Roche, Amgen, Takeda, Pierre Fabre, Sun Pharma, Sanofi outside the submitted work. J.G. is consultant of MSD, Novartis, Pfizer and Bayer. J.F.T. is a Scientific Advisory Board member for

GlaxoSmithKline, BMS, MSD Australia, and Provectus Inc. R.A.S. is a Scientific Advisor, Board Member for Merck Sharp & Dhome (MSD) and Novartis. None of these relationships involve the work described in this manuscript. The remaining authors declare no competing interests.

