## [Peer Review File · Nature Communications]

Reviewers' comments:

Reviewer #1 (Remarks to the Author):

1. Despite the significantly larger sample size analysed in this manuscript, all the recurrently mutated genes and driver oncogenes, structural variants and rearrangements for mucosal melanoma were previously reported by the same authors (Hayward, et al. Nature 2017). Several other findings were also previously reported by Furney, et al. Journal of Pathology, 2013. It seems a larger sample size was still not sufficient to identify any new driver gene.
2. The discussion contains several hypotheses based on the presented genomic landscape of mucosal melanoma, however, these seem a bit superficial. For example, based on mutation profile, the authors suggest a potential susceptibility of mucosal melanoma to CDK4/6 inhibitors alone or in combination with MEK inhibitors or immunotherapy. The authors should provide experimental evidence for these therapy regimens.
3. The manuscript would be significantly improved if, in addition to the descriptions of the mutational profiles, these could be studied for their clinical/prognostic implications.
4. Fig 3a and 3c suggest mucosal melanoma at nasal site doesn't have any BRAF mutation. Authors have not discussed such findings. Also, authors should perform a detailed investigation on whether different genetic background (Chinese vs Caucasian) has any impact on the genomic landscape of mucosal melanoma.

Reviewer #2 (Remarks to the Author):

This is an extension of a previous analysis of mucosal melanomas from some of the same authors, with a larger sample size that provided the opportunity to explore differences by body sites and other features. Although still a small study, and not original, it is the largest to date, given the rarity of this tumor subtype, and would be of interest to the melanoma and cancer research community.

I have several comments or questions.

The abstract is misleading. Only 67 melanomas underwent whole genome sequencing and provided estimates of mutations, telomere length, copy number alterations and structural variants. The remaining 45 FFPE samples underwent whole exome sequencing and were used only as a validation for driver gene mutations.

Based on the data in Supplemental Table 1, only 28 tissue samples were primary melanomas, the remaining were recurrences, lymph nodes metastases or had unknown status. This is important and needs to be stated, because the mutations and structural variants identified may reflect the clones that allowed cell migration to other sites but not the full characteristics of the primary tumors. And the mutational signature analysis could also be different in primary vs. metastatic samples. An analysis stratifying the main results between primary and metastatic/recurrence sites would be important. For example, mutations/loss in BAP1 or ATRX have been associated with increased risk of metastasis across different cancer types. Were the samples carrying these mutations metastatic/recurrent mucosal melanomas?

It is known that sample purity based on histological assessment often poorly reflect the actual tumor content. What was the sample purity based on copy number alterations (and variant allele fraction in the case of copy neutral samples)? Apparent lack of specific driver mutations or other changes may be due to low sample purity. This is particularly important for the primary melanomas, which are likely to be very small and the samples may be contaminated by the

surrounding tissue

The finding of UV-related signature 7 in 6 samples is intriguing. Are these primary melanoma samples? I am wondering whether these melanomas that could have been driven by the accumulation of UV-related mutations show different patterns of driver genes or structural variants in comparison to the other mucosal melanomas. Can a description of these 6 vs the other samples be reported? This could suggest different patterns of tumor initiation between the two groups.

Still related to mutational signatures: the contribution of signature 1 was significantly more prevalent in melanoma from lower body sites than upper body sites. What was the estimated power to distinguish signatures based on the sample size of upper and lower body sites? What were the mutational signatures in the FFPE samples? What were the 'normal', reference samples for the FFPE tumor tissue?

Whole genome sequencing was carried out in three different centers. More details are needed to ascertain that the approaches, e.g., filtering process for mutation calling, were consistent across the centers.

Was the telomere length analysis adjusted for age? Also, the associations between TERT mutations and short telomeres, and ATRX mutations with longer telomere have already been reported and investigated in detail across multiple cancer types (Barthel, Nature Genetics 2017), thus they may not need to be reported in the abstract.

Except for four samples, all melanomas included in the WGS analysis were from China and Australia. Is there any difference between these melanomas arising from subjects with different pigmentation background (besides the body site distribution)?

The ascertainment of structural variants is known to be challenging as spurious rearrangements are common. Can at least a percentage of the SVs be validated in the lab?

"Some evidence" of chromotripsis is reported. What was the evidence based on? As the analysis of chromotripsis in PCAWG shows (reported in BioRxiv), there are specific criteria and tests to define chromotripsis.

SPRED1 and NF1 were reported as almost mutually exclusive, since 11/13 mutations in SPRED1 were in NF wild type tumors. However, SPRED1 mutations were identified in 6 (based on WGS) and 1 (based on WES) tumors only. How many of these were NF1 wild type?

Whole-genome landscape of mucosal melanoma reveals diverse drivers and therapeutic targets (NCOMMS-18-36512A)

Reviewers' questions and comments are in black, bold text and the Authors' responses are in blue text. Manuscript additions in response to the reviewer's comments are outlined below and some additional text in the manuscript (primarily from the Discussion, as highlighted in the manuscript) was also removed in order to meet the word count requirements of *Nature Communications*.

Reviewer 1

- 1. Despite the significantly larger sample size analysed in this manuscript, all the recurrently mutated genes and driver oncogenes, structural variants and rearrangements for mucosal melanoma were previously reported by the same authors (Hayward, et al. Nature 2017). Several other findings were also previously reported by Furney, et al. Journal of Pathology, 2013. It seems a larger sample size was still not sufficient to identify any new driver gene.**

We thank the reviewer for highlighting our previous work but the larger sample size analysed here did indeed identify new driver genes for mucosal melanoma: *SPRED1* (previously unreported at the time of submission of this manuscript), *ATRX* and *CTNNB1*. In addition, the use of whole genome sequencing on this large cohort of samples identified novel recurrent structural rearrangements in major melanoma genes *TERT*, *CDK4* and *MDM2*, which therefore can also be considered drivers of mucosal melanoma.

- 2. The discussion contains several hypotheses based on the presented genomic landscape of mucosal melanoma, however, these seem a bit superficial. For example, based on mutation profile, the authors suggest a potential susceptibility of mucosal melanoma to CDK4/6 inhibitors alone or in combination with MEK inhibitors or immunotherapy. The authors should provide experimental evidence for these therapy regimens.**

We have re-worded the relevant paragraph in the Discussion to be more circumspect (see below) and now cite a very recent publication (ahead of print while our manuscript was under review) showing in PDX models that the majority of mucosal melanomas do indeed respond well to CDK4 inhibitors. This experimentally supports our conclusion. We were not in a position to conduct such studies ourselves since neither cell lines nor PDX models were available or generated from the tumors utilised in our study.

- Page 19, revised text: Driver events affecting CDK4 in a large proportion of our cases points to the potential therapeutic option of using CDK4 inhibitors to treat this tumor type. Indeed, Zhou et al ⁴⁸ have very recently shown that such inhibitors are effective in treating patient derived xenografts of mucosal melanomas carrying CDK4 aberrations, thus providing experimental support for this notion.

- 3. The manuscript would be significantly improved if, in addition to the descriptions of the mutational profiles, these could be studied for their clinical/prognostic implications.**

We agree that an in-depth analysis for clinical/prognostic implications would add another important layer of analysis to our study. Whilst we have tried to perform additional analysis for associations of clinical/outcome and genomic data, the clinical data was insufficient and

limited the conclusions that could be drawn. For survival analysis, the follow-up times for the Australian and Chinese cohorts were very different, with the follow-up time for the China patients being very short, (under 2 years for all patients) with a median follow-up time of 4.4 months and only one death (from melanoma). Meaningful survival data would thus be limited to just the Australian samples reducing the cohort size (n=24) and the power to detect associations. Furthermore, of these 24 patients, 5 died from unknown causes and another had unknown vital status, thus limiting power even further. Patients also received different treatments (this is now included in Supplementary Data 1) increasing the complexity of such analysis. A number of patients received chemotherapy (n=15, all Chinese) and/or immunotherapy/small molecular inhibitors (n=5, all Australian) after sample collection. We have summarized the survival and treatment data now in Supplementary Data 1 and have indicated its limitations for analysis in the text.

- Page 5 added text: Survival and treatment data are also included in Supplementary Data 1, however due to short follow-up times in the samples from China (all under 2 years, median of 4.4 months), associations of survival with genomic features were not formally analysed.

We have, however, made new several observations within the manuscript about associations with age. We have noted that samples of East Asian ancestry have diagnosis of primary mucosal melanoma at a younger age (page 6) and a lower proportion of the age-related mutation signature 1 (page 7). We also previously noted that mucosal melanomas with linking translocations on chr5 and chr12 have a lower age of diagnosis of the primary tumor.

4. Fig 3a and 3c suggest mucosal melanoma at nasal site doesn't have any BRAF mutation. Authors have not discussed such findings.

We have now highlighted these findings in the manuscript, along with the occurrence of one non-V600 BRAF mutation (G469A) arising from a nasal primary (1/17 WGS nasal samples):

- Page 10, added text: *BRAF* mutations were rare in the nasal cavity, with no codon 600 mutations and only one G-loop mutation identified (G469A).
- Page 15, added text in bold: For example, *SF3B1* hotspot mutations are common in anorectal and vulvovaginal melanomas but are rare in mucosal melanomas from other sites^{3, 28} **and *BRAF* mutations are less common in the nasal cavity.**

Due to the clinical relevance of this question, we have now added Supplementary Figure 5 (see below) to aid in the interpretation of the identified driver mutation frequency in the various primary sites.

Also, authors should perform a detailed investigation on whether different genetic background (Chinese vs Caucasian) has any impact on the genomic landscape of mucosal melanoma.

In response to this suggestion and a similar query (question 9) from Reviewer 2, we have performed an analysis of Australian versus Chinese samples, and updated the manuscript accordingly. It should be noted however that relatively low sample numbers and uneven body site distribution of tumors from these two geographical regions reduces power and potentially confounds meaningful assessment, as 19/24 Australian samples were from lower body mucosal sites whereas only 7/39 of the Chinese samples were from lower body mucosal sites.

To make more meaningful comparisons with respect to genetic background, we used principal component analysis (PCA) to assess the genetic ancestry of each sample by comparing genotypes of the mucosal patients with samples from different populations examined in the 1000 genomes (1000G) project. These data have been included in a new figure (Supplementary Figure 1) – which is shown below.

All Chinese samples, and one Australian sample clustered with the East Asian ancestry population (ie China, Japan, Vietnam). Twenty-four samples clustered as having European ancestry. Three remaining Australian samples had an ancestry that was not European or East Asian. Comparing tumors of European (24) versus Chinese/Asian ancestry (40), most samples Chinese/Asian ancestry samples were from upper body mucosal sites (31 upper, 8 lower, 1 unknown) whilst European ancestry samples were mostly from lower body mucosal sites (16 lower, 5 upper body sites, 3 unknown). We made subsequent comparisons based on genetic ancestry (24 European vs 40 East Asian ancestry).

The text has been amended as follows:

- All main figures have been updated to include tracks showing sample origin (ie where the sample was collected and sample ancestry as determined by PCA)
- On page 5/6, added text: Ethnicity of patients was determined by using principal component analysis to compare the genotypes of WGS samples with 1000G phase III samples of known populations (Supplementary Figure 1). All Chinese samples and one Australian sample (n=40) clustered with the East Asian ancestry super population. Twenty-four samples clustered as having European ancestry and three remaining Australian samples had an ancestry that was not European or East Asian. Genetic ancestry was used for subsequent comparisons between samples in terms of genomic features in this study. Based on genetic ancestry, the age at diagnosis of mucosal melanoma was younger in patients of East Asian ancestry, compared with patients of European ancestry (P=0.01, mean age 53 vs 63).
- In Methods on page 24: a section has now been added with the heading “Determination of sample genetic ancestry” to describe the methods used to determine genetic background.

When examining mutation signatures, all of the samples with >50% UV were from China and signature 17 (any proportion) was also found only in samples from China (5 upper body site and 1 lower body site). The proportion of signature 1 (Age) was higher in European samples than in Asian samples. Chinese samples had a lower average age at the development of the primary mucosal melanoma than samples of European ancestry (European mean=63, Asian mean=53, P=0.01). This difference in age may have contributed towards the difference observed between upper and lower body sites in terms of the proportion of mutation signature 1. There was no association between SV group 1 and 2

(from Figure 2, $P=0.24$), but for samples with chr5-chr12 translocations, 7 of 8 were from China and all were oral mucosal melanomas. Relative telomere length was shorter in European samples ($P=0.013$). In the manuscript (see Figure 5), it was observed that shorter relative telomere length was associated with lower body mucosal sites and the majority of European samples were from the lower body mucosal sites.

The text has been amended as follows:

- Page 7, revised text in bold: ...all of which were from upper mucosal body sites (**and of Chinese origin**), except one from an unknown primary site (Fig. 1a,c).
- Page 7, added text: Signature 17, of unknown aetiology, was present only in samples ($N=6$) of Chinese ancestry and mostly (5/6) in upper body mucosal sites.
- Page 7, added text: However, lower body site tumors in this study are predominantly from European patients and there was a significant difference in the age at diagnosis between patients of European and Asian ancestry (mean age 63 vs 53, $P=0.01$)
- Page 8, added text in bold: The relative proportion of Group 1 and Group 2 tumors did not differ between upper and lower body location (Fisher's exact $P=0.45$), **or by sample genetic ancestry (Fisher's exact $P=0.24$)**.
- Page 9, added text in bold: Most of the samples with chromosome 5p-12q translocations were oral mucosal melanomas (7 oral, 1 anorectal), **of East Asian ancestry (7 East Asian, 1 European)**, ...
- Page 9, modified text in bold: When analysing oral mucosal samples alone, samples with chr5p-12q translocations also had a lower age at diagnosis (**$P=0.008$, mean age 39 vs 58 any ancestry, $P=0.01$ samples of East Asian ancestry only**), but no difference by gender.
- Page 13, added text in bold: Lower body site ($P=0.0022$) **and tumors from patients of European ancestry ($P=0.013$)** were also associated with reduced telomere length (Supplementary Fig 7g,h).
- Discussion, page 15 added text in bold: We show different mutation signatures (based on UVR-related and endogenous mutagenic processes) occur in mucosal melanomas arising in facial sites compared to those arising in lower body sites and **and mutational signatures 7 and 17 occur more often in patients of East Asian ancestry**.
- Discussion, page 15 added text in bold: Together, our results demonstrate that mucosal melanomas show considerable heterogeneity based on the underlying mutagenic processes and body site-specific driver mutations **and that genetic ancestry or geographic location may also be factors associated with this**.
- Discussion, page 15 added text: The fact that UV signatures occur predominantly in samples of East Asian ancestry may be because mostly upper body site tumors in this study were from Chinese patients, or may reflect other geographic factors. Likewise, the fact that signature 17 occurred only in patients of East Asian ancestry also indicates geographically specific environmental or genetic factors may play a role in the development of mucosal melanomas. A larger cohort of samples from each ethnicity would be required to fully elucidate these factors.
- Discussion page 16 add text: While there was no significant difference in age of diagnosis between the patients with upper and lower body mucosal melanomas, patients of East Asian ancestry, which made up the greater proportion of upper body site tumors were, on average, younger than patients of European ancestry. Therefore, further work is required to establish if the presence of more signature 1 in the lower body site tumors reflects real site specific biology or is a result of the ethnicity of the patients whose tumor samples were examined in our study.

With respect to statistical associations of the genomic aberrations, there was no difference in the presence of mutations except for the genes that are mentioned in the manuscript amendments outlined below:

- Page 10, added text: *SF3B1* mutations also were mostly in mucosal melanomas of European ancestry (7/8).
- Page 11, added text in bold: There were no other relationships evident between SMGs and primary melanoma anatomic site, primary compared with metastatic/recurrent site or patient ancestry.
- Page 12, added text: The only gene where CNV aberrations was associated with ancestry was *NOTCH2* amplifications, with 4/6 aberrations being in European tumors.
- Page 14, added text: When comparing the profile of mutations of mucosal melanomas from East Asian and European ancestry, there were no differences in the presence of driver mutations in any particular genes, with the exception of *SF3B1* which was predominant in patients of European ancestry (Fisher's exact, P=0.011).
- Discussion, page 18, added text in bold: All but one of the eight tumors in this study with translocations between 5p and 12p, usually resulting in amplifications of *MDM2/CDK4* and *TERT*, were oral mucosal melanomas **and of East Asian ancestry**.

Reviewer 2

1. **This is an extension of a previous analysis of mucosal melanomas from some of the same authors, with a larger sample size that provided the opportunity to explore differences by body sites and other features. Although still a small study, and not original, it is the largest to date, given the rarity of this tumor subtype, and would be of interest to the melanoma and cancer research community.**

We thank the reviewer for his/her positive comments about our manuscript and viewpoint that it will be of interest to the melanoma and cancer research community.

2. **I have several comments or questions.**

The abstract is misleading. Only 67 melanomas underwent whole genome sequencing and provided estimates of mutations, telomere length, copy number alterations and structural variants. The remaining 45 FFPE samples underwent whole exome sequencing and were used only as a validation for driver gene mutations.

The Abstract has been modified to clarify the numbers of specimens that underwent whole genome sequencing and validation whole exome sequencing, respectively.

- Revised text: To better understand the genomic landscape of mucosal melanoma, here we describe whole genome sequencing analysis of 67 tumors and validation of driver gene mutations by exome sequencing of 45 tumors.

3. **Based on the data in Supplemental Table 1, only 28 tissue samples were primary melanomas, the remaining were recurrences, lymph nodes metastases or had unknown status. This is important and needs to be stated, because the mutations and structural variants identified may reflect the clones that allowed cell migration to other sites but not the full characteristics of the primary tumors. And the mutational signature analysis could also be different in primary vs. metastatic samples.**

We have added a sentence (see below) to the Results to highlight which samples are primary (n=31) or recurrent/metastatic tumors (n=26) and have performed an analysis

between these groups (excluding those of unknown specimen site). There was a higher number of SNV/indels in tumors from recurrent or metastatic sites (P=0.033) but no difference in the overall number structural variants, percent of the genome affected by CNV, telomere length or mutation signatures. We have amended the text as follows to clarify these issues:

- Page 5, added text: Samples were from primary tumors (n=31) as well as recurrent and metastatic sites (n=26), with 10 of unknown type.
- Page 6, added text: Tumor samples that were primary tumors had a significantly lower mutation load than samples from recurrent/metastatic tumors (P=0.033).

An analysis stratifying the main results between primary and metastatic/recurrence sites would be important. For example, mutations/loss in BAP1 or ATRX have been associated with increased risk of metastasis across different cancer types. Were the samples carrying these mutations metastatic/recurrent mucosal melanomas?

As *BAP1* (SNV, N=2) or *ATRX* (any aberration, N=6) mutated samples were in only a small percentage of the samples, it is therefore difficult to make meaningful comparisons. We did not observe an association with metastasis for *BAP1*, with a single mutation each in a primary and recurrent/metastatic sample. *ATRX* aberrations were not statistically associated with primary or metastatic samples (P=0.17).

With respect to other aberrations highlighted in the manuscript (ie SNV/indel, CNV or SV individually or all aberrations considered together for drivers listed in Figure 6), there were no statistically significant differences in the presence of mutations at primary versus recurrent/metastatic sites except:

Gene	Mutation type	# primary	# recurrent/metastatic	P-value
SF3B1	SNV/indel	8	0	0.0057
SF3B1	Any aberration	8	0	0.0057
NRAS	SNV/indel	2	8	0.032
NRAS	Any aberration	3	10	0.013
SPRED1	Any aberration	11	2	0.024

(all numbers above exclude samples of unknown sample site)

The text has been amended to include the observations/analyses highlighted above:

- Page 10, added text: *SF3B1* mutations also were mostly in tumors of European ancestry (7/8) and were all from primary tumor samples.
- Page 10, added text: *NRAS* mutations were also mostly found in recurrent/metastatic tumors (2 primary, 8 recurrent/metastatic, 2 unknown, Fisher's exact P=0.032).
- Page 14, added text: When comparing any aberration with respect to primary and recurrent/metastatic sample types, *SF3B1* (Fisher's exact, P=0.0057) and *SPRED1* (Fisher's exact, P=0.024) mutations were mostly present in primary tumors and *NRAS* (Fisher's exact, P=0.013) aberrations were predominantly in recurrent/metastatic tumors.

4. It is known that sample purity based on histological assessment often poorly reflect the actual tumor content. What was the sample purity based on copy number alterations (and variant allele fraction in the case of copy neutral samples)? Apparent lack of specific driver mutations or other changes may be due to low sample purity. This is particularly important for the primary melanomas, which are likely to be very small and the samples may be contaminated by the surrounding tissue

Tumor purity was derived from the WGS data using ascatNgs, a tool which uses copy number alterations to estimate tumor content. Where ascatNGS was unreliable after manual review, mean variant allele frequency was used. Purity for each tumor is listed in Supplementary Data 1. There was no significant association between tumor mutation burden and tumor purity (P=0.95). We thank the reviewer for identifying this issue and we have added the below to the Methods section to clarify this point:

- Added text, page 21: All tumors had a minimum tumor purity of 15%. Tumor purity was assessed using ascatNGS. Where ascatNGS was unreliable (following manual review), mean variant allele frequency was used (Supplementary Data 1).

5. The finding of UV-related signature 7 in 6 samples is intriguing. Are these primary melanoma samples? I am wondering whether these melanomas that could have been driven by the accumulation of UV-related mutations show different patterns of driver genes or structural variants in comparison to the other mucosal melanomas. Can a description of these 6 vs the other samples be reported? This could suggest different patterns of tumor initiation between the two groups.

The six samples that had >50% UV signature were samples from China, and two were primary melanoma samples (2 were metastatic/recurrent samples and 2 of unknown type). With respect to overall features, there were no significant differences in number of SNV/indels, number of SVs, percent of the genome affected by CNV or relative telomere length in tumors with >50% UV signature and less than 50% UV signature. We have now incorporated these findings in the text:

- Page 7, added text: There were no significant differences in the number of SNV/indel, structural variants or percent of the genome affected by copy number aberrations between samples with >50% UV signature and <50% UV signature.

With respect to mutations, there were no statistical differences in the presence of SNV/indel in driver genes (Figure 3) CNV and SV mutations (Figure 4) or aberrations of any mutation type shown in Figure 6. Such statistical differences would be difficult to detect however, given the imbalance in sample numbers between groups (i.e. 6 vs 61). The tumors with >50% UV contribution did however, lack mutations in: *TP53*, *SPRED1*, *SF3B1*, *PTEN*, *MITF*. We have now incorporated this statement in the text.

- Page 10 added text: The six tumors with > 50% UV signature had no statistically significant differences in driver gene mutations, but lacked mutations in *TP53*, *SPRED1*, *SF3B1*.

6. Still related to mutational signatures: the contribution of signature 1 was significantly more prevalent in melanoma from lower body sites than upper body sites. What was the estimated power to distinguish signatures based on the sample size of upper and lower body sites?

As the p-value when comparing mutation signature 1 in lower and upper body sites is <0.001 and the Mann-Whitney test statistic is approximately Gaussian for these sample sizes, it follows that the observed power is >99.95%. In one of the publications that first described mutational signature analysis, Alexandrov and co-workers (DOI: 10.1016/j.celrep.2012.12.008) performed an estimation of the ability of the technique to accurately identify signatures. They found that with mutations from a total of 50 simulated

cancer genomes, extracting 7 or more signatures requires an average of at least 1000 mutations per genome. In our cohort, we performed analysis with 67 samples and with the number of mutations per sample had a mean of 7737 mutations (range 1563-20593), with 7 signatures identified, which meet the minimum criteria to accurately identify this number of signatures. In terms of the ability to identify specific differences between upper and lower body sites in the current study, there are 24 lower site tumors and 36 upper site tumors and due to whole genome sequencing, there are approximately 400-4700 mutations per sample assigned to signature 1, which represents a sufficient number of mutations to allow comparisons between body sites. It should be noted however, based on the additional analyses outlined elsewhere in this response to reviewer comments, that the observed differences between lower and upper body sites may be confounded by a range of parameters including age (which was notably different between China and Australian patients: the proportion of patients from European and East Asian ancestry is different between upper & lower body sites). We have therefore modified the discussion of signature 1 in the manuscript to reflect the additional analyses performed in our response to the reviewer comments:

- Page 16, added text: While there was no significant difference in age of diagnosis between the patients with upper and lower body mucosal melanomas, patients of East Asian ancestry, which made up the greater proportion of upper body site tumors were, on average, younger than patients of European ancestry. Therefore, further work is required to establish if the presence of more signature 1 in the lower body site reflects real site specific biology or is a result of the ethnicity of the patients whose tumors samples were examined.

What were the mutational signatures in the FFPE samples? What were the ‘normal’, reference samples for the FFPE tumor tissue?

The normal reference samples were matched normal tissues for the site of each tumor. These details have been added to the table in Supplementary Data 2. With respect to mutational signatures in exome sequencing data from FFPE samples, it should be noted that assignment of mutational signatures is likely to be unreliable due to the very low numbers of SNVs. A cut-off of 100 SNVs to assign signatures in FFPE samples has previously been applied (Wong et al, 2019 <https://doi.org/10.1038/s41467-018-08081-1>) and only 7 of the FFPE samples described here met this cut-off. The mutational signatures in the 7 FFPE samples with 100 or more SNV mutations were assigned to the seven signatures identified through WGS and were found to be predominantly signature 1, as well as some contribution of signatures 5 or 3-like in several samples. One sample from the nasal cavity had a UV signature contribution of 30%. Two other samples had small UV signature contributions (12 and 15%, but given the low number of mutations in the samples (representing only about 12-30 mutations assigned as UV signature), this assignment is likely to be unreliable. As we consider that this does not add any significant new information to the manuscript, we have not added the described analysis to the manuscript, but have added the following text on page 25: Mutational signatures were not analysed in the FFPE validation cohort due to low mutation numbers.

7. Whole genome sequencing was carried out in three different centers. More details are needed to ascertain that the approaches, e.g., filtering process for mutation calling, were consistent across the centers.

We appreciate the reviewer highlighting that this section in the Methods was unclear. Although sequencing was carried out at three centres, all downstream analysis, including

alignment, variant calling and filtering were carried out at one site, QIMR Berghofer using the same pipeline (as previously described in Hayward et al, 2017 <https://doi.org/10.1038/nature22071>). To make this clearer, we have modified the methods section to include the following on page 21:

“All downstream processing, including sequence alignment and variant calling was carried out at QIMR Berghofer (Brisbane, Australia) using the same analysis pipeline for all WGS samples.”

- 8. Was the telomere length analysis adjusted for age? Also, the associations between TERT mutations and short telomeres, and ATRX mutations with longer telomere have already been reported and investigated in detail across multiple cancer types (Barthel, Nature Genetics 2017), thus they may not need to be reported in the abstract.**

The telomere length analysis was not adjusted for age since such an adjustment is not required here, as the measurement we used involves tumor telomere length compared to the matched normal from each patient. A normalised telomere count was obtained for each tumor bam and its matching normal bam by counting reads with telomeric repeats and then normalising this count to the average genomic coverage in the bam. The relative telomere length is a ratio of normalized tumor counts to normal counts. Shorter or longer telomeres are with respect to the normal telomere length for that particular individual, and it is therefore not necessary to adjust for age. Although, associations between telomere length and *ATRX* and *TERT* have been previously reported, these have not been described in mucosal melanoma, therefore we feel that this observation warrants inclusion in the Abstract.

- 9. Except for four samples, all melanomas included in the WGS analysis were from China and Australia. Is there any difference between these melanomas arising from subjects with different pigmentation background (besides the body site distribution)?**

Please see the answer to Reviewer 1, question 4 where we have addressed this question in detail.

- 10. The ascertainment of structural variants is known to be challenging as spurious rearrangements are common. Can at least a percentage of the SVs be validated in the lab?**

The qSV tool has been used in previous publications (Patch et al, 2015, Nature 521(7553):489-94. doi: 10.1038/nature14410; Waddell et al, 2015 518(7540):495-501. doi: 10.1038/nature14169) where SVs were validated by identifying the same variants using orthogonal (SOLiD) sequencing. Unfortunately, due to low amounts of available DNA for samples, SV validation of these samples in the lab was not possible. Importantly, the structural rearrangement tool we have used (qSV) uses multiple lines of evidence to reduce false positives (soft clipping, discordant pairs and split reads). During the development of the qSV tool, extensive validation of SV variants was performed. Some of this work is described in Quek et al (2014, Biotechniques, 51(1):31-38, doi: 10.2144/000114189), which discussed a workflow to validate structural variations. In that study, 356 SVs were tested by Sanger, MiSeq or IonTorrent sequencing, 248 (69.7%) were validated and a further 66 (18.5%) did not produce an amplicon, however, a negative result may not necessarily indicate that an event is false, as some PCRs may require individual optimisation in order to verify an event.

11. “Some evidence” of chromothripsis is reported. What was the evidence based on? As the analysis of chromothripsis in PCAWG shows (reported in BioRxiv), there are specific criteria and tests to define chromothripsis.

The evidence for chromothripsis and BFB are discussed in the Results on page 9 with the text:

“Although there was some evidence of chromothripsis (clustered breakpoints, oscillation of copy number and retention of heterozygosity) and breakage-fusion-bridge (loss of telomeric regions and high number of inversions), most events were too complex to confidently assign to one particular type of mutational mechanism.”

We have modified the text in the Results and Discussion for clarity:

- Results revised text, page 9: A review of chromosomes showing evidence of clustered breakpoints revealed the localised events had some features of genomic catastrophes. Although there were some tumors that showed patterns similar to chromothripsis (clustered breakpoints, oscillation of copy number and retention of heterozygosity) and breakage-fusion-bridge (loss of telomeric regions and a high number of inversions), most events were too complex to confidently assign to one particular type of mutational mechanism.
- Discussion, page 17 revised text: Regions of localised complexity could be the result of genomic catastrophes such as chromothripsis. We identified tumors that showed patterns similar to chromothripsis, as well as breakage-fusion-bridge (BFB).

12. SPRED1 and NF1 were reported as almost mutually exclusive, since 11/13 mutations in SPRED1 were in NF wild type tumors. However, SPRED1 mutations were identified in 6 (based on WGS) and 1 (based on WES) tumors only. How many of these were NF1 wild type?

Thank you for highlighting this information; the sentence was incorrectly worded. The proportion 11/13 refers to samples with any type of *SPRED1* mutation, including SNV, CNV and SV. When looking at SNV/indel mutations only, all *SPRED1* mutations (5 in WGS and 1 in WES) were *NF1* wildtype. We have therefore changed this sentence to clarify the statement:

Original text, page 16:point mutations (11/13) occurred in *NF1* wild-type samples, and only SVs in *SPRED1* co-occurred with *NF1* mutations.

Revised text:aberrations identified in WGS samples (11/13) occurred in *NF1* wild-type samples, with only two SVs in *SPRED1* co-occurring with *NF1* mutations.

REVIEWERS' COMMENTS:

Reviewer #1 (Remarks to the Author):

The manuscript may appear to lack originality and novelty, but this is a rare disease and the authors are complemented on the resource they committed completing to this study, and on providing such an extensive and complete bioinformatic analysis. This will provide deep insight into our understanding of mucosal melanoma genomics, and the manuscript will serve as an excellent resource for the mucosal melanoma community in particular, but also to the wider melanoma community in general. The manuscript therefore makes a major contribution to the field that will be much appreciated and so should be published in Nature Communications.

Richard Marais.

Reviewer #2 (Remarks to the Author):

The authors responded adequately to all my questions, conducted a lot of extra work and revised the manuscript accordingly. I have no additional comments.

REVIEWERS' COMMENTS:

Reviewer #1 (Remarks to the Author):

The manuscript may appear to lack originality and novelty, but this is a rare disease and the authors are complemented on the resource they committed completing to this study, and on providing such an extensive and complete bioinformatic analysis. This will provide deep insight into our understanding of mucosal melanoma genomics, and the manuscript will serve as an excellent resource for the mucosal melanoma community in particular, but also to the wider melanoma community in general. The manuscript therefore makes a major contribution to the field that will be much appreciated and so should be published in Nature Communications.

Richard Marais.

Reviewer #2 (Remarks to the Author):

The authors responded adequately to all my questions, conducted a lot of extra work and revised the manuscript accordingly. I have no additional comments.

We thank the reviewers for reviewing the manuscript and for their constructive comments during the review process